# Evidence suggesting that reindeer mothers allonurse according to the direct reciprocity and generalized reciprocity decision rules

Sacha C. Engelhardt [1,2,3,4]*, Robert B. Weladji[3], Øystein Holand[5], Knut H. Røed[6], Mauri Nieminen[7]

**1** Department of Sociobiology/Anthropology, Johann-Friedrich-Blumenbach Institute for Zoology und Anthropology, University of Göttingen, Göttingen, Lower Saxony, Germany, **2** Behavioral Ecology and Sociobiology Unit, German Primate Center, Leibniz Institute for Primate Research, Göttingen, Lower Saxony, Germany, **3** Department of Biology, Concordia University, Montreal, Quebec, Canada, **4** Department of Biology, Institute of Ecology and Evolution, University of Bern, Hinterkappelen, Bern, Switzerland, **5** Department of Animal and Aquacultural Sciences, Norwegian University of Life Sciences, Ås, Viken, Norway, **6** Department of Preclinical Sciences and Pathology, Norwegian University of Life Sciences, Oslo, Oslo, Norway, **7** Natural Resources Institute Finland Luke, Reindeer Research Station, Kaamanen, Lapland, Finland

\* sacha.engelhardt@uni-goettingen.de

**Data Availability Statement:** All relevant data are within the manuscript and its Supporting Information files. We included 8 csv files and 1 .RData file. The .RData file can be opened in R or Rstudio with the function: load("Data8.RData"). If

## Abstract

Allonursing is the nursing of the offspring of other mothers. Cooperation is an emergent property of evolved decision rules. Cooperation can be explained by at least three evolved decision rules: 1) direct reciprocity, i.e. help someone who previously helped you, 2) kin discrimination, i.e. preferentially direct help to kin than to non-kin, and 3) generalized reciprocity, i.e. help anyone if helped by someone. We assessed if semi-domesticated reindeer, *Rangifer tarandus*, mothers allonursed according to the decision rules of direct reciprocity, generalized reciprocity and kin discrimination over 2 years. To assess if reindeer mothers allonursed according to the direct reciprocity decision rule, we predicted that mothers should give more help to those who previously helped them more often. To assess if reindeer mothers allonursed according to the kin discrimination decision rule, we predicted that help given should increase as pairwise genetic relatedness increased. To assess if reindeer mothers allonursed according to the generalized reciprocity decision rule, we predicted that the overall number of help given by reindeer mothers should increase as the overall number of help received by reindeer mothers increased. The number of help given i) increased as the number of help received from the same partner increased in the 2012 group but not in both 2013 groups, ii) was not influenced by relatedness, and iii) was not influenced by an interaction between the number of help received from the same partner and relatedness. iv) The overall number of help given increased as the overall number of help received increased. The results did not support the prediction that reindeer mothers allonursed according to the kin discrimination decision rule. The results suggest that reindeer mothers may allonurse according to the direct reciprocity and generalized reciprocity decision rules.

you have a problem downloading the file, please contact the corresponding author. The Rscript is included and can be read in R or Rstudio.

**Funding:** This project was supported by a doctoral bursary from the Fonds de recherche du Québec, Nature et technologies, a Northern Scientific Training Program award, and a QCBS Excellence Award granted to SCE. This project was also funded by RB's Natural Sciences and Engineering Research Council of Canada (https://www.nserc-crsng.gc.ca/index_eng.asp) Discovery Grant number 327505. The contributions of KR and ØH were funded by Reindeer Husbandry in a Globalizing North (ReiGN), which is a Nordforsk-funded (https://www.nordforsk.org/nordic-centre-excellence) "Nordic Centre of Excellence" (project number 76915). The funders had no role in study design, data collection and analysis, decision to publish, or preparation of the manuscript.

**Competing interests:** The authors have declared that no competing interests exist.

## Introduction

In cooperative and communal breeding social systems, members of the social group provide parental care to the offspring of other parents [1,2]. Cooperative breeding can be defined as breeding females assisted in protecting and caring for their offspring by non-breeding helpers, and communal breeding can be defined as breeding females that pool their young and share care and provisioning [3]. In singular breeding social systems, a single pair is responsible for all reproduction and subordinate group members do not produce young, whereas in plural breeding social systems, most or all adults within a group produce young of their own [4,5]. There are different types of alloparental care in cooperative and communal breeding social systems, such as communal defense of young, babysitting, adoption and allonursing. Allonursing is the provision of milk to the offspring of other mothers [6,7].

Lactation is the most energetically expensive aspect of mammalian reproduction and increases the energetic costs and metabolic demands, resulting in weight loss of lactating females [8,9]. In the first few days or weeks after parturition, milk is the exclusive source of nutrients for new-born mammals [10]. Allonursing increases nursing loads, and females with heavier nursing loads may incur greater risks of mortality [11] and lower future fecundity [11,12] than those with lighter nursing loads. Allonursing may increase the risk of pathogen transmission between offspring and mothers [13] and may decrease the amounts of nutrients available to a mother's offspring.

Allonursing occurs in over 68 mammalian species and across most mammalian families [6,7,14]. Animals in captivity allonurse more often than animals in natural populations, and animals with larger litter sizes allonurse more often than animals with smaller litters [6]. Allonursing is more common in polytocous species, i.e. giving birth to more than one offspring per parturition, than monotocous species, i.e. giving birth to one offspring per parturition [6,15]. Monotocous species tend to allonurse more often when group size is large, while polytocous species tend to allonurse more often when group size is small [6]. The occurrence of allonursing varies between taxa [6]. The incidence of allonursing is not greater in singular cooperative breeders, which have greater mean within-group relatedness, than in non-singular cooperative breeders, and the incidence of allonursing is not associated with relatedness in these groups [15].

The misdirected parental care, i.e. either due to milk-theft and mismothering [7], kin selection [7], reciprocity [7], compensation [16], improved nutrition [6,17], milk evacuation [7], neuroendocrine [18] and immunological function [13] hypotheses were proposed to explain why females allonurse the offspring of other mothers. The kin selection hypothesis proposes that lactating females preferentially allonurse closely over distantly related offspring, providing inclusive fitness benefits [7]. The reciprocity hypothesis proposes that two females achieve a higher fitness when nursing each other's offspring to a similar extent than when they do not share milk [7]. Reciprocal allonursing is the reciprocated feeding of offspring.

Altruism is a behaviour that is costly to the actor and beneficial to the recipient [19]. Helping is a behaviour that provides an apparent benefit to the recipient [20]. Cooperation is the simultaneous or consecutive acting together of two or more individuals by same or different behaviours [20,21]. Kin selection is selection acting on the consequences of an individual's behaviour on the survival and/or reproduction of its relatives [22,23], and its three mechanisms are kin discrimination, limited dispersal and greenbeards [19,24]. Hamilton's rule sums up indirect fitness and asserts that altruistic traits can spread within populations if the product of the degree of relatedness, $r$, and the recipient's benefit, $b$, is greater than the actor's costs, $c$ [19]. The indirect fitness benefits as a result of helping appear to be smaller than originally expected or may be underestimated, and the direct fitness benefits as a result of help appear to

be greater than originally expected and may even be greater than the indirect fitness benefits [21,25,26]. Both direct and indirect fitness can interact [21]. Help between kin is well explained by indirect fitness [27], which cannot explain help between non-kin (e.g. reciprocity) [20,21,25]. Reciprocity is an apparently cooperative trait or behaviour, which benefits the recipient of the help at a cost to the actor, that increases the probability of the actor of the helpful behaviour receiving help in return from the same or different social partners [20,21]. For reciprocity, the helping costs for the actor should be low; benefits for the receiver should be high; and the probability of receiving help in return for help given should be high [28]. Cooperation between unrelated individuals is common across a wide range of taxa [20,21,28]. Reciprocal interactions can also occur between related and unrelated individuals, such that the correlated pay-offs for the interacting partners can interact with relatedness [21], e.g. in female Norway rats, *Rattus norvegicus* [29].

Lehman and Keller [30] proposed a theoretical framework and classification of models for the evolution of cooperation. These models can be divided into four distinct categories [20]. Cooperation can arise from the net fitness benefits of i) by-product mutualism [25,31–34]. Cooperation can evolve from the net fitness benefits of altruism according to the correlated pay-offs of conditional returns due to ii) an above-random chance that help provided to a social partner will increase the likelihood of receiving help in return in the future, i.e. reciprocity [21,28,35,36]; or by the correlated pay-offs of shared genes iii) kin selection [19,24,37–39]. Cooperation can evolve by manipulation according to enforcement [40,41] or deception [42–46].

Cooperation is an emergent property of evolved decision rules [19–21,24,28,35,47–50]. Cooperation can be explained by at least three evolved, mechanistic decision rules: 1) direct reciprocity, i.e. help someone who previously helped you (if A helped B, B helps A) [20,21], 2) kin discrimination is the differential treatment of conspecifics as a function of their genetic relatedness to the actor [20,24], and at the proximate mechanism level it implies a decision rule: preferentially direct help to kin than to non-kin (if A and B are kin, but A and C are non-kin, A preferentially helps B), and 3) generalized reciprocity, i.e. help anyone if helped by someone (if A helped B, B is more likely to help anyone) [20,21]. These decision rules can generate evolutionarily stable cooperation [19–21,24,28,35,47–49]. Kin discrimination requires that individuals can discriminate relatives from non-relatives and occurs by the use of environmental or genetic cues [51]. Direct reciprocity requires individual recognition and the ability to remember the outcomes of past interactions with an individual [21,52,53]. Generalized reciprocity only requires the ability to remember if one received help or not in previous social interactions, without needing to remember and identify the partner(s) [20,21].

Reciprocal cooperation has been reported in vertebrates and invertebrates [20,21,54,55]. Direct reciprocity was reported across several taxa, including mammals [28,29,56–62] (e.g. vampire bats, *Desmodus rotundus* [57,58] and Norway rats, *Rattus norvegicus* [29,59–61,63–65]), birds [20,21], fishes [54,55], and microorganisms [66,67]. Generalized reciprocity was reported in humans [68,69], female Norway rats [60,64,70], dogs, *Canis familiaris* [71], capuchin monkeys, *Sapajus apella* [72], however generalized reciprocity was not supported in male Norway rats [64,73], long-tailed macaques, *Macaca fascicularis* [74] and in vampire bats [57]. Generalized reciprocity is also known as upstream tit-for-tat [47], upstream indirect reciprocity [75], upstream reciprocity [76], pay it forward [77–79], and serial reciprocity [80].

Most of the evidence for the importance of kin selection for the evolution of cooperation is correlational. For empirical studies in taxa with a high social complexity, i.e. eusocial and cooperative breeding social systems, the association between relatedness and cooperation was positive [81–84], negative [85–87] or neutral [88–94]. In a comparative analysis of cooperatively breeding vertebrates, helping rates increased positively with relatedness, which explained 10% of the variation in helping rates [95], whereas altruism declined with increasing

intragroup relatedness in a comparative analysis of eusocial hymenoptera [96]. Several studies that experimentally manipulated relatedness found that relatedness reduced cooperation rather than increase it [29,87,97,98], which contrasts with kin selection theory. Cooperation can be explained by reciprocity and relatedness, and the fitness pay-offs of the interacting partners are correlated [20,21]. Theoretical models predict i) a difference [99], and ii) an interaction in the contribution of direct reciprocity and relatedness for the evolution of cooperation [100,101]. Cooperation was better explained by direct reciprocity than relatedness [29,57,102] and generalized reciprocity [74] in some empirical studies. Direct reciprocity and generalized reciprocity can co-exist, such as in humans [68], female Norway rats [60,64,70], and in dogs [71,103]. Theoretically, cooperation usually stems from the interaction of multiple evolutionary mechanisms [99–101,104,105]. Very little is known about the relative importance of these 3 decision rules, except in male Norway rats [29,73].

Previous findings suggested that reindeer, *Rangifer tarandus*, mothers allonursed i) reciprocally [106], ii) to improve the nutrition and mass gain of offspring [107], and iii) by kin discrimination [97], whereas offspring were allonursed by stealing milk [108]. Reindeer mothers reciprocated allonursing at the group level, i.e. across bouts and dyads, and most mothers had at least one reciprocal partner [106]. Allonursing by reindeer mothers was positively associated with offspring mass gain, which is consistent with improved nutrition, and male offspring gained more mass than female offspring [107]. Relatedness did not influence the odds of allonursing [108], however a group of closely-related mothers allonursed each other's offspring more often than a group of distantly-related mothers at the extremes of pairwise genetic relatedness [97]. There is no evidence in reindeer to support the mismothering [108], the compensation [107] and the milk evacuation [109] allonursing hypotheses. There is indirect evidence suggesting that reindeer can discriminate kin. Reindeer mothers rejected offspring attempting to be allonursed more often than they rejected their own offspring attempting to be nursed [108]. Reindeer mothers rejected nearly all attempts by offspring to be allonursed before the mother's offspring was suckling [97,108], and most attempts by offspring to be allonursed occurred after the mother's offspring was suckling [108].

Most allonursing studies reported no evidence to support reciprocal allonursing [6,7]. There are few empirical studies of the generalized reciprocity decision rule in animals [60,64,68,70–72,74], and we need to further our knowledge of how widespread the generalized reciprocity decision rule is in animals. Most animal species should be able to apply the generalized reciprocity decision rule, since several theoretical models found that generalized reciprocity can generate evolutionarily stable levels of cooperation [21,47,48,49,75–77,110–114], and the decision rule of generalized reciprocity is less cognitively demanding, i.e. a simpler decision rule, than the direct reciprocity decision rule [20,21]. No study of allonursing and, more generally, alloparental care has yet tested the generalized reciprocity decision rule nor compared it with the direct reciprocity decision rule.

There is an on-going debate over the utility of kin selection and alternative mechanisms [20,21,51,115–119]. This reveals the need to experimentally test combinations of evolved decision rules to further understand the interactions and relative importance of the mechanisms responsible for the evolution of cooperation. To improve on the superficial appreciation of the involved mechanisms responsible for the evolution of cooperation, i) we address the evolved, mechanistic decision rules of direct reciprocity, generalized reciprocity and kin discrimination responsible for cooperation, and ii) we combine evolved decision rules of cooperation to assess the interaction between the direct reciprocity and kin discrimination decision rules.

To further our understanding of the decision rules of reciprocal and kin-discriminated allonursing, we tested if semi-domesticated reindeer mothers allonursed according to the decision rules of direct reciprocity, generalized reciprocity and kin discrimination. To assess if reindeer

mothers allonursed according to the direct reciprocity decision rule, we predicted that 1) mothers should give more help to the offspring of mothers who previously helped them more often, and 2) the likelihood to allonurse should increase as pairs of mothers are more reciprocal. To assess if reindeer mothers allonursed according to the kin discrimination decision rule we predicted that 3) mothers should give more help to offspring as the pairwise genetic relatedness of mothers increased, 4) the likelihood to allonurse should increase as the pairwise genetic relatedness of mothers increases, and 5) the extent of reciprocity within pairs of mothers should increase as the pairwise genetic relatedness of mothers increases. To assess if reindeer mothers allonursed according to an interaction between the direct reciprocity and kin discrimination decision rules, we predicted that 6) the effect of help received on help given should depend on the relatedness between pairs of mothers, and 7) the effect of the extent of reciprocation within pairs of mothers on the likelihood of allonursing should depend on the relatedness between pairs of mothers. To assess if reindeer mothers allonursed according to the generalized reciprocity decision rule, we asked if receiving help in general increased subsequent help given, and we predicted that 8) the overall number of help given by reindeer mothers should increase as the overall number of help received by reindeer mothers increased. We compared reciprocal allonursing in reindeer by both the generalized reciprocity and direct reciprocity decision rules. We propose that the propensity to help should be greater according to direct reciprocity than according to generalized reciprocity, since direct reciprocity is less prone to cheating than generalized reciprocity. Therefore, the effects of received help should vanish more quickly if generalized reciprocity applies, so we predicted that 9) the distribution of latencies to give help after receiving help should peak at shorter time intervals for generalized reciprocity than for direct reciprocity.

## Materials and methods

This study was carried out in strict accordance with the Animal Ethics and Care Certificate of Concordia University. The protocol was approved by the Animal Ethics and Care Certificate of Concordia University (Protocol number: AREC-2010-WELA). We conducted this study at the Kutuharju Field Reindeer Research Station near Kaamanen, Finland (69˚ N, 27˚ E): a 45 km$^2$ fenced enclosure. The Kutuharju Field Reindeer Research Station is a research station, and the reindeer are owned by reindeer herders, who manage the size of groups and in which sections of the station different groups will roam during the calving season. The herd is also managed to avoid inbreeding. Yearly, the semi-domesticated female reindeer are herded into open fenced paddocks (approximately 10 ha), where data on the birth date, calf sex, and mother-calf assignments were obtained. This population has been monitored since 1969 [120]. Female reindeer are monotocous, giving birth to one offspring in May–June, and they are plural breeders. Mothers gradually wean their offspring, and the lactation cycle usually ends in September–October during rut [121].

In 2012, the first 25 offspring born and their mothers were selected and studied for 65 observations days over 10 weeks [106–108]. In 2012, the 25 offspring were born between May 4th and May 13th. In 2013, we studied two groups for 25 observations over five weeks and each group consisted of eight mother-offspring pairs based on their pairwise genetic relatedness, forming two groups at the extremes of pairwise genetic relatedness: one group of closely, genetically related mothers and one group of distantly, genetically related mothers [97]. We manipulated relatedness in 2013 by choosing reindeer mothers to form 2 groups at the extremes of pairwise genetic relatedness in the population to assess if relatedness at the extremes of relatedness in the population affected help allonursing given [97]. In 2013, the 16 offspring were born between May 8th and May 23rd. Mother-offspring pairs were assigned

within 24–48 hours of parturition. No mother was removed from their own mother when they themselves were young. Mothers were not separated from their offspring. Thus, reindeer mothers' ability to distinguish kin from non-kin was not affected. Mothers gave birth to their offspring and raised their offspring with the rest of the herd in the calving paddocks for two to 5 weeks. Mother-offspring pairs selected for the study were then separated from the rest of the herd, which was released in a large calving ground area. The study animals and the herd were rejoined at the end of the study. Researchers fixed collar tags of different colours, with numbers inscribed, to individuals for identification. The manager of the herd maintained a population of females ranging between 2–13 yrs in 2012 (mean ± SD = 7.32 ± 3.21 yrs) and between 2–12 yrs in 2013 (mean ± SD = 5.93 ± 3.76 yrs). The age of mothers ranged from 2–13 yrs in the 2012 group (mean ± SD = 8.25 ± 2.96 yrs). The age of mothers ranged from 3–8 yrs in the 2013 closely-related group study group (mean ± SD = 6.00 ± 2.51 yrs), and the age of mothers ranged from 5–11 yrs in the 2013 distantly-related group (mean ± SD = 8.12 ± 2.42 yrs). Detailed descriptions of methods used to collect data in 2012 and 2013 were previously published [97,106,108].

A successful allonursing bout was scored when an offspring was nursed for 5 s or more and ended when the offspring no longer suckled the lactating female's udder. A successful allonursing bout was considered terminated when an offspring's muzzle and a lactating female's udder were not in contact for 20 s or more. We selected a 5 s cut-off based on previous research [108,122–125]. An unsuccessful allonursing attempt was scored when i) an offspring brought its muzzle within a head from a lactating female's udder, which did not allow the offspring the suckle (e.g. walking away, kicking calf, head threat to calf, chasing calf), or ii) the offspring suckled for less than 5 s [108,124], and the termination was due to the lactating female not allowing the offspring to suckle any longer. Previous studies of allonursing in reindeer reported that most mothers allonursed and most offspring were allonursed [122,126]. In 2012, all reindeer mothers allonursed one or more offspring, and we recorded 1383 allonursing bouts [108]. In the closely and distantly, genetically related groups in 2013, we recorded 113 and 48 allonursing bouts, respectively, performed by eight of the eight reindeer mothers in the closely-related group and by seven of the eight reindeer mothers in the distantly-related groups [97]. In both groups in 2013, seven of the eight offspring were allonursed [97]. Across both years, we recorded 1544 allonursing bouts. It was not possible to record data blind because our study involved focal animals in the field. In this study, we revisited the associations between help given and i) help received and ii) pairwise genetic relatedness and their interactions by updating the assessment of reciprocal and kin-discriminated allonursing by including all data collected in 2012, i.e. all allonursing bouts lasting > 5 s rather than only the data with exact known duration, and by adding a 2nd year of data, i.e. data collected in 2013. This allowed us to further our understanding of the evolved, mechanistic decision rules responsible for reciprocal allonursing.

All mothers were habituated to human presence. Observations were conducted inside the paddock at a distance ranging from 5 to 50 meters from animals by 3 trained observers in 2012 and 1 trained observer in 2013. Binoculars were used to reliably record observations of solicitations, agonistic interactions, and identify individuals. Observations of nursing and allonursing solicitations were collected on data collection sheets using behaviour sampling with continuous recording [127]. For each solicitation, the occurrence of nursing and allonursing and the identity of the mother and offspring were recorded. A solicitation was scored as an attempt when an offspring brought its muzzle within a head from a mother's udder, and the mother did not allow the offspring to be nursed (e.g. walking away, kicking, head threat, chasing). A solicitation was scored as a rejection when the offspring suckled for less than 5 seconds [124], and the termination was due to the mother not allowing the offspring to be nursed any longer.

A solicitation was scored as successful (i.e. a bout) when an offspring suckled for 5 seconds or more and ended when the offspring no longer grasped the mother's udder. We selected a 5 seconds cut-off based on previous nursing and allonursing research with reindeer [122,128], and a 5 seconds cut-off has also been selected to study nursing and allonursing in red deer, *Cervus elaphus* [16,129], fallow deer, *Dama dama* [130], cows, *Bos taurus* [124], bactrian camels (*Camelus bactrianus*) [125], zebras, *Equus greyvi* [131], bighorn sheep, *Ovis canadensis* [132].

Four behaviour sampling sessions of agonistic interactions of one hour each, using continuous recording, were conducted daily in 2012 [127]. Throughout each observation day in both 2012 and 2013, agonistic interactions were opportunistically scored using ad libitum sampling and continuous recording methods. An agonistic interaction was recorded as resolved when an individual showed a submissive behaviour ("lose"), and the other did not ("win"). Unresolved agonistic interactions were recorded as unresolved when neither animal showed a submissive behaviour. The agonistic interactions scored were displacement, head threat, push, chase, kick, boxing, and other interactions [133,134], and their associated submissive behaviours were scored as 'flee' or 'walk away' if submission occurred. The rank of female reindeer is fairly stable throughout the year [135], except for a very short time immediately following the shedding of antlers [134–136].

We collected blood samples from all individuals and analyzed for 16 DNA microsattelite loci as part of an on-going progeny testing within this experimental herd [137]. We assessed parenthood assignments with the simulation program software CERVUS 3.0 [138], which is based on likelihood ratios between candidate parents. We found all microsatellites within the herd to be in Hardy Weinberg equilibrium and detected no mismatches in the assigned mother-offspring combinations. The DNA analyses supported all mother-offspring assignments from field observations. We used the program GenAlEx v 6.4 [139] to generate methods-of-moments estimators of pairwise genetic relatedness, LRM [140]. In 2013, we selected groups at the extremes of pairwise genetic relatedness, and we selected one group of eight mothers to be closely, genetically related and the other group of eight mothers to be distantly, genetically related [97]. In 2013, we selected study animals for both groups with similar birth masses, age of offspring, age of mothers and the numbers of male and female offspring [97]. The LRM estimates of pairwise genetic relatedness were calculated for the 3 groups (2012: mean ± SE = -0.009 ± 0.003, 95% CI = -0.016–0.002, range = -0.144–0.239; 2013 closely-related: mean ± SE = 0.024 ± 0.011, 95% CI = 0.00059–0.047, range = -0.072–0.152; 2013 distantly-related: mean ± SE = -0.023 ± 0.007, 95% CI = -0.038 –-0.009, range = -0.071–0.062).

## Statistical analyses

There were 3 observers in 2012 and 1 observer in 2013. Inter-observer reliability was assessed in 2012 between the pairs of observers using Pearson correlation coefficients for the duration of allonursing, and the index of concordance for the identities of mothers and offspring allonursing [127].

Observations of successful allonursing across bouts were summed within dyads for the closely and distantly, genetically related groups. To assess dyadic reciprocity, we used the reciprocity index presented by Mitani [141], which has been used to assess dyadic reciprocity for allogrooming [141,142] and for allonursing [106], to create the reciprocal allonursing frequency index (RAFI).

$$RAFI = 1 - \left| \frac{aAB}{aA + aB} - \frac{aBA}{aA + aB} \right| \tag{1}$$

Where $aAB$ is the amount of allonursing that individual $A$ gave to individual $B$'s offspring,

*aBA* is the amount of allonursing that individual *B* gave to individual *A*'s offspring, and *aA* + *aB* is the total amount of allonursing between the two individuals. The reciprocity index quantifies the degree to which members of a dyad match one another's exchange of allonursing bouts. The reciprocal indexes range from 0 (no reciprocation and unidirectional, i.e. allonursing performed by only one individual) to 1 (complete reciprocation). Index values equal to or above 0.5 were interpreted as a tendency towards reciprocity; values below 0.5 were interpreted as a tendency towards unidirectionality; values of 0.8 or above were interpreted as strong reciprocity [106,142].

We recorded 4471 resolved, i.e. a winner and a loser were identified, agonistic interactions between mothers in 2012 [106]. We recorded 517 resolved agonistic interactions between mothers in the closely related group and 627 agonistic interactions between mothers in the distantly-related group in 2013 [97]. We generated a dominance hierarchy for reindeer mothers in each group using observations of agonistic interactions with a winner and loser identified. We reported the Landau linearity index [143]. The dominance hierarchy in the closely-related group tended to be linear, with a Landau linearity index of 0.857 [97]. The dominance hierarchy in the distantly-related group was linear, with a Landau linearity index of 0.988 [97]. In 2012, the dominance hierarchy tended to be linear, with a Landau's index of linearity of 0.785 [106]. Ranks were given values ranging 1 to 25 in 2012 and ranging 1 to 8 in both groups in 2013, with the largest number representing the most dominant mother and 1 representing the least dominant mother. Female primates exchange allogrooming for itself and for rank-related benefits, and they allogroom reciprocally [62,144–146]. We included absolute rank difference in the statistical models for the direct reciprocity and kin discrimination decision rules to assess if reindeer mothers allonurse for rank-related benefits.

To assess differences in the age of offspring between groups, we ran a generalized linear mixed model with a Poisson distribution with age of the offspring on May 30[th] in either 2012 or 2013 as the response and group as the fixed effect. Year was a random intercept effect. Post hoc Tukey comparisons were performed for multiple comparisons (i.e. distantly-related vs closely-related groups), and we corrected the alpha to 0.025 to account for multiple testing.

Following a recommendation by Kline [147], all continuous variables were standardized to z-scores to detect outliers with absolute z-score values greater than 3.3, and we used the scale function to calculate the z-scores. If individuals with outlier scores or influential scores were not from the same population as the rest of the individuals, then it may have been best to remove that case from the sample [147]. We assessed for influential values with dfbetas and Cook's d. The cutoff values for dfbetas and Cook's were based on $2/\sqrt{N}$ and $4/N$, respectively, with N representing the number of mothers (2012: N = 25; both groups in 2013: N = 8). The cutoff values for dfbetas were 0.40 in the 2012 group and 0.71 in both of the 2013 groups, and the cutoff values for Cook's d were 0.16 in the 2012 group and 0.50 in both of the 2013 groups. The individuals did belong to the population, and it was not best to remove these individuals or values from the sample. The influential values were real data points from the study population, so we kept them to avoid generating biased results due to selective removal of outliers [147–149].

To assess the predictions for the direct reciprocity decision rule (prediction 1), the kin discrimination decision rule (prediction 3) and the interaction between the direct reciprocity and kin discrimination decision rules (prediction 6), we ran one generalized linear mixed model for each group with a Poisson distribution with the number of help given as a response variable, and the interaction between the number of help received and the pairwise genetic relatedness, the absolute rank difference, the offspring birth mass difference, the similarity in offspring sex (categorical variable with 2 levels: same sexes = 1, different sexes = 0) as fixed effects. The offspring birth mass difference and the similarity in offspring sex were control

variables. The individual identities of both mothers in each dyad were used as random effects. If the interaction term between the number of help and pairwise genetic relatedness was not significant (i.e. there was no interaction between the direct reciprocity and kin discrimination decision rules; prediction 6), we removed the interaction term to estimate the main effect of the number of received help to test the prediction for the direct reciprocity decision rule (prediction 1) and the main effect of pairwise genetic relatedness to test the prediction for the kin discrimination decision rule (prediction 3). If the interaction term was not significant, we could use the conditional main effects to assess the predictions for the direct reciprocity decision rule (prediction 1) and the kin discrimination decision rule (prediction 3). To further assess the relationship between help received and help given, a generalized linear mixed model with a Poisson distribution with the same fixed and random effects was conducted with dyads with a tendency to reciprocate, i.e. 95 dyads with RAFI values equal or greater than 0.50, in the 2012 group with the main effects for the number of help received and for pairwise genetic relatedness. The residuals were not overdispersed in most models, except for the 2013 distantly-related group with the main effects for the number of help received and for pairwise genetic relatedness. To account for overdispersion, a generalized linear mixed model with a negative binomial distribution and the same fixed and random effects was conducted, and the residuals were not overdispersed. We tested for multicollinearity using the "vif" function from the car package. Variation inflation factor values of $\geq 5.00$ are interpreted as evidence of multicollinearity [150]. There was no multicollinearity in the models, except for the statistical model for the 2013 distantly-related group with the interaction term between the number of help received and pairwise genetic relatedness (the interaction and the conditional main effect of the number of help received both had variation inflation factor values > 130). By removing the interaction term in this model, there was no multicollinearity. In the 2012 group, there were 4 outliers for the number of help received, and 2 outliers for pairwise genetic relatedness. In the closely-related group in 2013, there was 1 outlier for the number of help received. In the distantly-related group in 2013, there was 1 outlier for the number of help received. There were 3, 1, 0, 3, 1, 4 influential values in the models for the 2012 group with the interaction, for the 2012 group with the main effects for the number of help received and pairwise genetic relatedness, for the 2012 group's dyads with a tendency to reciprocate (RAFI $\geq 0.50$) with the main effects for the number of help received and pairwise genetic relatedness, for the 2013 closely-related group with the interaction, for the 2013 closely-related group with the main effects for the number of help received and pairwise genetic relatedness, for the 2013 distantly-related with the main effects for the number of help received and pairwise genetic relatedness, respectively. Model results with the corrected outlier values were similar to the model results with the outliers, so we reported the model results with the outliers. We reported incidence rate ratios, their 95% confidence intervals and conditional $R^2$.

To further assess the predictions for the direct reciprocity decision rule (prediction 2), the kin discrimination decision rule (prediction 4) and the interaction between the direct reciprocity and the kin discrimination decision rules (prediction 7), we ran separate generalized linear mixed models per study group with a binomial distribution with successful (1) and unsuccessful (0) allonursing attempts as a response variable, and RAFI, pairwise genetic relatedness, offspring sex (categorical variable: male vs female), and absolute rank difference as fixed effects. The response variable of the likelihood of allonursing model equals 0 when an attempt is unsuccessful, i.e. the lactating female did not allonurse the offspring of another mother, and equals 1 when an attempt is successful, i.e. the lactating female did allonurse the offspring of another mother. The number of help received and the number of help given by mothers accounts for successful allonursing bouts received and given by mothers without accounting for the allonursing rejections by mothers, i.e. unsuccessful allonursing attempts,

however the likelihood of allonursing model accounts for both the successful and unsuccessful allonursing attempts. We assessed whether more reciprocal dyads of mothers may be more likely to allonurse, i.e. the odds of a successful allonursing attempt may be greater than the odds of an unsuccessful allonursing attempt for more reciprocal dyads. In the 2012 data, there were 11 outliers for pairwise genetic relatedness with an absolute z-score values greater than 3.3. The outliers were real data points, so we kept them. There were no outliers in the 2013 data. To assess the impact of these outliers, we changed them to the highest value below an absolute z-score of 3.3, and we ran the model with these corrected outlier values. Model results with the corrected outlier values were similar to the model results with the outliers, so we reported the model results with the outliers. There was no multicollinearity. For each model, when the interaction between pairwise genetic relatedness and RAFI was not significant, the interaction was removed from the model to assess the main effects of RAFI and pairwise genetic relatedness to test predictions 2 and 4. The individual identities of both individuals in each dyad were used as random intercept effects. For the 2013 groups, the random intercept effects explain zero variance. When this occurs, the glmer function with binomial distribution returns the results of the glm function with a binomial distribution.

Two internal meta-analyses were performed to assess the direct reciprocity decision rule among the three groups of reindeer mothers with the inverse variance method of pooling to assess the pooled effect size with fixed effects models based on the between-study heterogeneity. The effect sizes were the incidence rate ratio and the odds ratio, and both effect sizes were derived from the generalized linear mixed models. The estimates of the effect sizes were comparable among groups, since they were generated from generalized linear mixed models with the same distributions, and fixed and random effects.

To further assess the prediction for the kin discrimination decision rule (prediction 5), we ran separate generalized linear mixed models per study group with either a Poisson distribution or a negative binomial distribution, if the residuals were overdispersed, with the direct reciprocity index, RAFI, as the response variable. The values of RAFI were multiplied by 100 and rounded to the nearest integer. The fixed effects were the pairwise genetic relatedness, the absolute rank difference of mothers, the difference in offspring birth mass, and the absolute difference in the age of mothers. The individual identities of both individuals in each dyad were used as random intercept effects. In the 2012 data, there were 2 outliers for pairwise genetic relatedness with an absolute z-score values greater than 3.3. There were no outliers in the 2013 data. All values were real data points from the population, so we kept them. To assess the impact of these outliers, we changed them to the highest value below an absolute z-score of 3.3, and we ran the model with these corrected outlier values. Model results with the corrected outlier values were similar to the model results with the outliers, so we reported the model results with the outliers. There was multicollinearity in the 2013 models and variables were removed.

To assess the prediction for the generalized reciprocity decision rule (prediction 8), we assessed if receiving help in general increased subsequent help given, and we ran a general linear model with a Poisson distribution with the overall number of help given by each reindeer mother as a response variable, and the overall number of help received by each reindeer mother as an independent variable. To account for the different number of observation days between years, an offset for the number of observation days was included. There were 2 random intercept effects: the identity of mothers receiving and giving help and year. There were two influential values and one outlier. The generalized reciprocity decision rule states help anyone if helped by someone. The cognitive demands for generalized reciprocity only require the ability to remember whether or not one received help without having to identify or remember the partner(s) [20,21]. The identity of the individual from whom one received or

did not receive help is not relevant and so is the identity of future partners. To compare the latency to give help to anyone after having receiving help (prediction 9), we calculated the observed latencies to give help to anyone after having received help. When the time at the end of a received help was not recorded, we added the mean duration of allonursing bouts, i.e. 15 s, to the time at the start of receiving help. This decision has no significant effect on the latencies to give help to anyone after having received help, since reindeer mothers i) did not simultaneously help each other, ii) rarely helped each other within the same day, and the latencies between help received and help given typically span multiple days. It is not possible to assign each allonursing bout given after having received help as direct reciprocity or generalized, so we calculated the latencies to give help after receiving help according to both direct reciprocity and generalized reciprocity. To compare direct reciprocity and generalized reciprocity, we ran a generalized linear mixed model with a Gaussian distribution with the latency to give help after receiving help in days as a response variable. The comparison between the direct reciprocity and generalized reciprocity decision rules as levels of a categorical variable, the age of mothers and offspring sex (M vs F), and pairwise genetic relatedness were the fixed effects. There were two random intercept effects, i.e. the identity of mothers receiving help and giving help in return and year. The latency to give help after receiving help in days was log-transformed, so the model residuals were normally distributed. For all models, coefficients are reported with standard errors (SE), and an alpha of 0.05 was adopted. We used the "lme4" [151], "MASS" [152], "lmerTest" [153], "ggplot2" [154], "effects" [155,156], "tidyverse" [157], "gridExtra" [158], "multcomp" [159], "MuMIn" [160], "performance" [161], "car" [156] and "extrafont" [162] R packages in Rstudio [163] with R version 4.2.2 [164].

## Results

The Pearson correlation coefficients for the duration of allonursing between the 3 pairs of observers were 0.997 (N = 418), 0.969 (N = 217), and 0.999 (N = 45). The identities of mothers and offspring allonursing were reliably measured between the 3 pairs of observers (indexes of concordance = 1.0). The number of nursing bouts per mother was 207.04 ± 7.95 (95% CI: 191.46–222.62) in the 2012 group, 49.25 ± 2.48 (95% CI: 44.39–54.11) in the closely-related group and 59.38 ± 7.89 (95% CI: 43.91–74.84) in the distantly-related group in 2013. The number of allonursing bouts per mother was 55.32 ± 4.02 (95% CI: 47.44–63.20) in the 2012 group, 14.13 ± 3.15 (95% CI: 7.95–20.30) in the closely-related group and 6.00 ± 2.12 (95% CI: 1.84–10.16) in the distantly-related group in 2013. In 2012, the mean duration of nursing bouts was 2.71 times longer than that for allonursing bouts (nursing bouts: 40.23 ± 0.63 s, 95% CI = 39.00–41.46 s, N = 3396 nursing bouts; allonursing bouts: 14.85 ± 0.29 s, 95% CI = 14.29–15.42 s, N = 1022 allonursing bouts). In 2013, the mean duration of allonursing bouts was 16.82 ± 0.74 s (95% CI = 15.37–18.27 s, N = 89 allonursing bouts).

The age of offspring on May 30th in the 3 groups was calculated (2012: mean ± SE = 22.08 ± 0.47, 95% CI = 21.11–23.05; 2013 closely-related: mean ± SE = 15.63 ± 0.86, 95% CI = 13.58–17.67; 2013 distantly-related: mean ± SE = 15.38 ± 1.84, 95% CI = 11.02–19.73). The offspring in the closely-related group were younger on May 30th 2013 than the offspring on May 30th 2012 ($\beta \pm SE$: -0.35 ± 0.10, $P < 0.001$), and the offspring in the distantly-related group were younger on May 30th 2013 than the offspring on May 30th 2012 ($\beta \pm SE$: -0.36 ± 0.10, $P < 0.001$). There was no difference in the age of offspring in the closely- and distantly-related groups on May 30th 2013 (Post-hoc Tukey distantly-related vs closely-related: Estimate ± $SE$: -0.02 ± 0.13, $P = 0.99$).

## Decision rules of direct reciprocity, kin discrimination and their interaction for the 2012 group

The interaction between the number of help received and pairwise genetic relatedness did not significantly affect the number of help given ($\beta \pm SE$: 0.29 ± 0.22, $P$ = 0.17, IRR (95% CI): 1.34 (0.88–2.06)) in the 2012 group. The conditional main effect for the number of received help when pairwise genetic relatedness was equal to 0 significantly influenced the number of help given ($\beta \pm SE$: 0.04 ± 0.02, $P$ = 0.017, IRR (95% CI): 1.05 (1.01–1.08)). The conditional main effect for the pairwise genetic relatedness when the number of received help was equal to 0 did not significantly influence the number of help given ($\beta \pm SE$: -0.17 ± 0.74, $P$ = 0.82, IRR (95% CI): 0.85 (0.19–3.60)). Absolute rank difference ($\beta \pm SE$: -0.0008 ± 0.0063, $P$ = 0.90, IRR (95% CI): 1.00 (0.99–1.01)), similarity in offspring sex ($\beta \pm SE$: 0.10 ± 0.07, $P$ = 0.13, IRR (95% CI): 1.10 (0.97–1.26)), and offspring birth mass difference ($\beta \pm SE$: -0.11 ± 0.08, $P$ = 0.19, IRR (95% CI): 0.90 (0.76–1.06)) did not significantly influence the number of help given. The intercept was significant ($\beta \pm SE$: 0.91 ± 0.15, $P$ < 0.001). The random intercept effect for the first mother in a dyad explained 0.06 (SD = 0.24) of the variance, and the random intercept effect for the other mother in a dyad explained 0.25 (SD = 0.50) of the variance. The model's conditional $R^2$ was equal to 0.52.

The number of help given increased as the number of help received increased (main effect: $\beta \pm SE$: 0.05 ± 0.02, $P$ = 0.0096, IRR (95% CI): 1.05 (1.01–1.09), Fig 1A) in the 2012 group. For a one unit increase in the number of help received, the number help given is expected to increase by a factor of 1.05 (95% CI for IRR: 1.01–1.09). If a mother received help 1, 10 or 15 times, the mother is expected to give help 2.61, 4.10 and 5.26 times, respectively. The number of help given was not significantly influenced by the pairwise genetic relatedness (main effect: $\beta \pm SE$: 0.39 ± 0.61, $P$ = 0.52, IRR (95% CI): 1.48 (0.44–4.89), Fig 1B). Absolute rank difference ($\beta \pm SE$: -0.002 ± 0.006, $P$ = 0.76, IRR (95% CI): 1.00 (0.99–1.01)), similarity in offspring sex ($\beta \pm SE$: 0.10 ± 0.07, $P$ = 0.13, IRR (95% CI): 1.10 (0.97–1.26)), and offspring birth mass difference ($\beta \pm SE$: -0.10 ± 0.08, $P$ = 0.24, IRR (95% CI): 0.91 (0.77–1.08)) did not significantly influence the number of help given. The intercept was significant ($\beta \pm SE$: 0.91 ± 0.15, $P$ < 0.001). The random intercept effect for the first mother in a dyad explained 0.06 (SD = 0.24) of the variance, and the random intercept effect for the other mother in a dyad explained 0.26 (SD = 0.51) of the variance. The model's conditional $R^2$ was equal to 0.52.

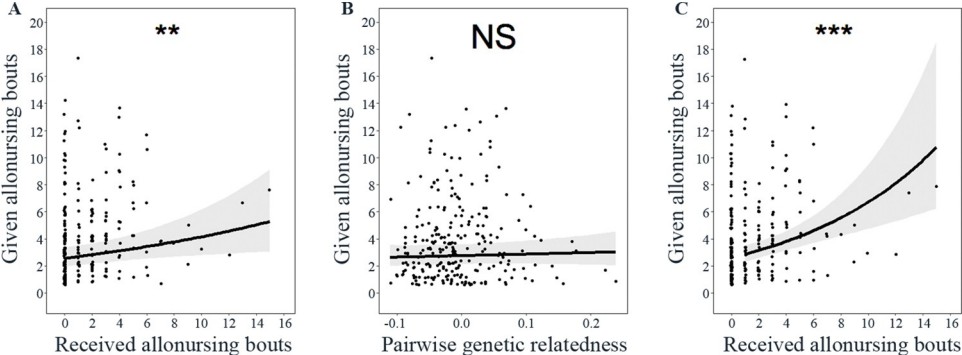

**Fig 1.** Help given and (A) the direct reciprocity and (B) the kin discrimination decision rules for all dyads and (C) the direct reciprocity decision rule for dyads with a tendency to reciprocate. Given allonursing bouts and received allonursing bouts represent the number of allonursing bouts given and the number of allonursing bouts received in the 2012 group. A and B) All dyads, i.e. RAFI ranging from 0.00 to 1.00. C) Only the dyads with a tendency to reciprocate, i.e. RAFI ranging from 0.50 to 1.00. The predicted effects (black line) are plotted with 95% confidence intervals (grey areas). The black dots represent the raw data of food donations. NS > 0.05; **$P$ < 0.01; ***$P$ < 0.001.

For the 95 dyads with a tendency to reciprocate (RAFI $\geq$ 0.50) in the 2012 group, the number of help given increased as the number of help received increased (main effect: $\beta \pm SE$: 0.09 $\pm$ 0.02, $P <$ 0.001, IRR (95% CI): 1.10 (1.05–1.15), Fig 1C). For a one unit increase in the number of help received, the number help given is expected to increase by a factor of 1.10 (95% CI for IRR: 1.05–1.15). If a mother received help 1, 10 or 15 times, the mother is expected to give help 2.48, 5.58 and 8.76 times, respectively. The number of help given was not significantly influenced by the pairwise genetic relatedness (main effect: $\beta \pm SE$: 1.45 $\pm$ 1.03, $P =$ 0.16, IRR (95% CI): 4.28 (0.54–32.24)). Absolute rank difference ($\beta \pm SE$: 0.008 $\pm$ 0.009, $P =$ 0.37, IRR (95% CI): 1.01 (0.99–1.03)), similarity in offspring sex ($\beta \pm SE$: 0.16 $\pm$ 0.11, $P =$ 0.13, IRR (95% CI): 1.18 (0.95–1.46)), and offspring birth mass difference ($\beta \pm SE$: -0.13 $\pm$ 0.07, $P =$ 0.08, IRR (95% CI): 0.88 (0.76–1.02)) did not significantly influence the number of help given. The intercept was significant ($\beta \pm SE$: 0.82 $\pm$ 0.16, $P <$ 0.001). The random intercept effect for the first mother in a dyad explained 3.97e-10 (SD = 1.99e-5) of the variance, and the random intercept effect for the other mother in a dyad explained 0.08 (SD = 0.29) of the variance. The model's conditional $R^2$ was equal to 0.38.

The interaction between RAFI and pairwise genetic relatedness did not significantly affect the likelihood to allonurse ($\beta \pm SE$: 0.56 $\pm$ 2.21, $P =$ 0.80, OR (95% CI): 1.75 (0.02–133.17)) in the 2012 group. The conditional main effect for RAFI when pairwise genetic relatedness was equal to 0 significantly influenced the likelihood to allonurse ($\beta \pm SE$: 0.72 $\pm$ 0.15, $P <$ 0.001, OR (95% CI): 2.05 (1.53–2.76)). The conditional main effect for the pairwise genetic relatedness when RAFI was equal to 0 did not significantly influence the number of help given ($\beta \pm SE$: 0.02 $\pm$ 1.20, $P =$ 0.99, OR (95% CI): 1.02 (0.10–10.72)). Sex of the offspring being allonursed (M vs F: $\beta \pm SE$: -0.23 $\pm$ 0.17, $P =$ 0.18, OR (95% CI): 0.79 (0.57–1.11)), absolute rank difference ($\beta \pm SE$: 0.003 $\pm$ 0.008, $P =$ 0.67, OR (95% CI): 1.00 (0.99–1.02)), and offspring birth mass difference ($\beta \pm SE$: -0.14 $\pm$ 0.08, $P =$ 0.09, OR (95% CI): 0.87 (0.74–1.02)) did not significantly influence the likelihood to allonurse. The intercept was not significant ($\beta \pm SE$: 0.08 $\pm$ 0.16, $P =$ 0.62). The random intercept effect for the allonursing mother in a dyad explained 0.05 (SD = 0.22) of the variance, and the random intercept effect for the mother of the offspring being allonursed in a dyad explained 0.09 (SD = 0.30) of the variance. The model's conditional $R^2$ was equal to 0.07.

The likelihood of allonursing increased as RAFI increased (main effect: $\beta \pm SE$: 0.71 $\pm$ 0.15, $P <$ 0.001, OR (95% CI): 2.03 (1.52–2.73), Fig 2A) in the 2012 group. The likelihood of allonursing was not significantly affected by the pairwise genetic relatedness (main effect: $\beta \pm SE$:

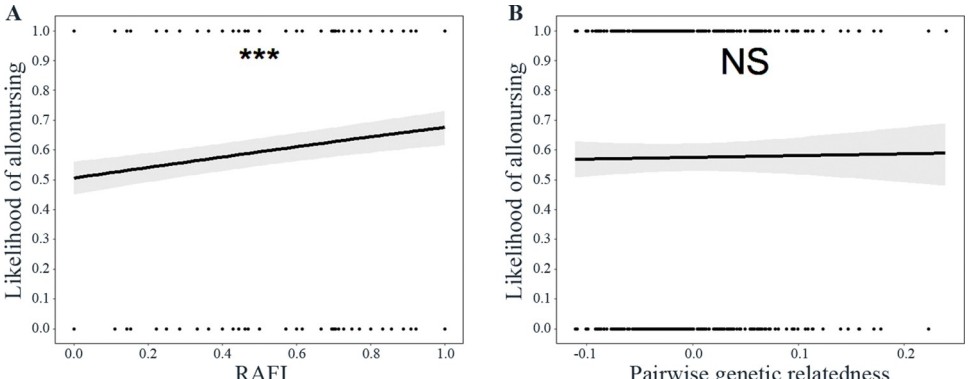

**Fig 2.** The likelihood of allonursing and the (A) direct reciprocity and (B) kin discrimination decision rules. RAFI represents the reciprocal allonursing frequency index values. The predicted effects (black line) are plotted with 95% confidence intervals (grey areas). The black dots represent the raw data of food donations. NS > 0.05, ***$P <$ 0.001.

0.24 ± 0.81, $P$ = 0.77, OR (95% CI): 1.27 (0.26–6.22), Fig 2B). Sex of the offspring being allonursed (M vs F: $\beta \pm SE$: -0.23 ± 0.17, $P$ = 0.18, OR (95% CI): 0.79 (0.57–1.11)), absolute rank difference ($\beta \pm SE$: 0.003 ± 0.008, $P$ = 0.66, OR (95% CI): 1.00 (0.99–1.02)) and offspring birth mass difference z-scored ($\beta \pm SE$: -0.14 ± 0.08, $P$ = 0.08, OR (95% CI): 0.87 (0.74–1.02)) did not significantly influence the likelihood of allonursing. The intercept was not significant ($\beta \pm SE$: 0.08 ± 0.16, $P$ = 0.61). The random intercept effect for the allonursing mother in a dyad explained 0.05 (SD = 0.22) of the variance, and the random intercept effect for the mother of the offspring being allonursed in a dyad explained 0.09 (SD = 0.30) of the variance. The model's conditional $R^2$ was equal to 0.07.

The pairwise genetic relatedness (main effect: $\beta \pm SE$: -0.81 ± 2.88, $P$ = 0.78, IRR (95% CI): 0.44 (1.58e-3–133.81)), the absolute rank difference ($\beta \pm SE$: 0.02 ± 0.03, $P$ = 0.47, IRR (95% CI): 1.02 (0.97–1.07)), offspring birth mass difference ($\beta \pm SE$: 0.02 ± 0.21, $P$ = 0.92, IRR (95% CI): 1.02 (0.67–1.52)), and the absolute age difference of mothers ($\beta \pm SE$: 0.02 ± 0.07, $P$ = 0.77, IRR (95% CI): 1.02 (0.89–1.17)) did not significantly influence RAFI in the 2012 group. The intercept was significant ($\beta \pm SE$: 2.22 ± 0.64, $P < 0.001$). The random intercept effect for the first mother in a dyad explained 6.50 (SD = 2.55) of the variance, and the random intercept effect for the other mother in a dyad explained 0.00 (SD = 0.00) of the variance. The model's conditional $R^2$ was equal to 0.62.

## Decision rules of direct reciprocity, kin discrimination and their interaction for the 2013 closely-related group

The number of help given in the 2013 closely-related group was not significantly influenced by the interaction between the number of help received and pairwise genetic relatedness ($\beta \pm SE$: 0.72 ± 1.22, $P$ = 0.56, IRR (95% CI): 2.05 (0.16–42.19)), the conditional main effect for the number of received help when pairwise genetic relatedness was equal to 0 ($\beta \pm SE$: -0.06 ± 0.07, $P$ = 0.41, IRR (95% CI): 0.94 (0.82–1.11)), the conditional main effect for the pairwise genetic relatedness when the number of received help was equal to 0 ($\beta \pm SE$: -4.77 ± 4.10, $P$ = 0.25, IRR (95% CI): 8.51e-3 (4.07e-7–23.41)), absolute rank difference ($\beta \pm SE$: 0.07 ± 0.10, $P$ = 0.51, IRR (95% CI): 1.07 (0.87–1.31)), similarity in offspring sex ($\beta \pm SE$: 0.46 ± 0.28, $P$ = 0.10, IRR (95% CI): 1.58 (0.84–2.75)), and offspring birth mass difference ($\beta \pm SE$: 0.41 ± 0.28, $P$ = 0.14, IRR (95% CI): 1.51 (0.78–2.66)). The intercept was not significant ($\beta \pm SE$: 0.59 ± 0.37, $P$ = 0.11). The random intercept effect for the first mother in a dyad explained 1.36e-8 (SD = 0.0001) of the variance, and the random intercept effect for the other mother in a dyad explained 0.23 (SD = 0.48) of the variance. The model's conditional $R^2$ was equal to 0.55.

The number of help given was not significantly influenced by the number of help received (main effect: $\beta \pm SE$: -0.05 ± 0.07, $P$ = 0.49, IRR (95% CI): 0.96 (0.83–1.10)) in the 2013 closely-related group. The number of help given was not significantly influenced by the pairwise genetic relatedness (main effect: $\beta \pm SE$: -3.43 ± 3.34, $P$ = 0.30, IRR (95% CI): 0.03 (2.97e-5–22.05)). Absolute rank difference ($\beta \pm SE$: 0.04 ± 0.09, $P$ = 0.66, IRR (95% CI): 1.04 (0.87–1.24)), similarity in offspring sex ($\beta \pm SE$: 0.44 ± 0.27, $P$ = 0.11, IRR (95% CI): 1.55 (0.88–2.69)), and offspring birth mass difference ($\beta \pm SE$: 0.41 ± 0.27, $P$ = 0.14, IRR (95% CI): 1.51 (0.86–2.62)) did not significantly influence the number of help given. The intercept was marginally significant ($\beta \pm SE$: 0.65 ± 0.35, $P$ = 0.06). The random intercept effect for the first mother in a dyad explained 1.10e-9 (SD = 3.31e-5) of the variance, and the random intercept effect for the other mother in a dyad explained 0.24 (SD = 0.49) of the variance. The model's conditional $R^2$ was equal to 0.54. There were 9 dyads with a tendency to reciprocate, i.e. RAFI $\geq$ 0.50.

The interaction between RAFI and pairwise genetic relatedness did not significantly affect the likelihood to allonurse ($\beta \pm SE$: -1.05 ± 7.63, $P$ = 0.89, OR (95% CI): 0.35 (1.11e-7–1.10e6) in the 2013 closely-related group. The conditional main effect for RAFI when pairwise genetic relatedness was equal to 0 did not significantly influence the likelihood to allonurse ($\beta \pm SE$: -0.34 ± 0.46, $P$ = 0.46, OR (95% CI): 0.71 (0.29–1.75)). The conditional main effect for the pairwise genetic relatedness when RAFI was equal to 0 did not significantly influence the number of help given ($\beta \pm SE$: -0.45 ± 4.66, $P$ = 0.92, OR (95% CI): 0.64 (6.88e-5–5.91e4)). Sex of the offspring being allonursursed (M vs F: $\beta \pm SE$: 0.53 ± 0.32, $P$ = 0.10, OR (95% CI): 1.70 (0.91–3.18)), absolute rank difference ($\beta \pm SE$: 0.09 ± 0.10, $P$ = 0.36, OR (95% CI): 1.09 (0.90–1.33)), and offspring birth mass difference ($\beta \pm SE$: -0.31 ± 0.20, $P$ = 0.12, OR (95% CI): 0.73 (0.50–1.09)) did not significantly influence the likelihood to allonurse. The intercept was not significant ($\beta \pm SE$: 0.11 ± 0.41, $P$ = 0.78). The random intercept effect for the allonursing mother in a dyad explained 0.00 (SD = 0.00) of the variance, and the random intercept effect for the mother of the offspring being allonursed in a dyad explained 0.00 (SD = 0.00) of the variance. The model's conditional $R^2$ was equal to 0.08.

The likelihood of allonursing was not significantly influenced by RAFI (main effect: $\beta \pm SE$: -0.36 ± 0.44, $P$ = 0.41, OR (95% CI): 0.70 (0.29–1.65), Fig 2A) in the 2013 closely-related group. The likelihood of allonursing was not significantly affected by the pairwise genetic relatedness (main effect: $\beta \pm SE$: -0.98 ± 2.71, $P$ = 0.72, OR (95% CI): 0.38 (0.002–76.07), Fig 2B). Offspring sex (M vs F: $\beta \pm SE$: 0.53 ± 0.32, $P$ = 0.10, OR (95% CI): 1.70 (0.91–3.18)), absolute rank difference ($\beta \pm SE$: 0.09 ± 0.10, $P$ = 0.35, OR (95% CI): 1.09 (0.90–1.33)) and offspring birth mass difference z-scored ($\beta \pm SE$: 0.31 ± 0.19, $P$ = 0.09, OR (95% CI): 0.73 (0.51–1.06)) did not significantly influence the likelihood of allonursing. The intercept was not significant ($\beta \pm SE$: 0.12 ± 0.40, $P$ = 0.76). The random intercept effect for the allonursing mother in a dyad explained 0.00 (SD = 0.00) of the variance, and the random intercept effect for the mother of the offspring being allonursed in a dyad explained 0.00 (SD = 0.00) of the variance. The model's conditional $R^2$ was equal to 0.08.

RAFI was not significantly influenced by the pairwise genetic relatedness (main effect: $\beta \pm SE$: -5.43 ± 12.80, $P$ = 0.67, IRR (95% CI): 0.004 (5.63e-14–3.43e+8)) in the 2013 closely-related group. RAFI decreased as the absolute age difference of mothers increased ($\beta \pm SE$: -0.75 ± 0.37, $P$ = 0.045, IRR (95% CI): 0.47 (0.23–0.98)). RAFI marginally decreased as offspring birth mass difference increased ($\beta \pm SE$: -2.37 ± 1.31, $P$ = 0.07, IRR (95% CI): 0.09 (0.007–1.22)). The intercept was marginally significant ($\beta \pm SE$: 3.42 ± 1.77, $P$ = 0.053). The random intercept effect for the first mother in a dyad explained 14.71 (SD = 3.84) of the variance, and the random intercept effect for the other mother in a dyad explained 0 (SD = 0) of the variance. The model's conditional $R^2$ was equal to 0.92.

## Decision rules of direct reciprocity, kin discrimination and their interaction for the 2013 distantly-related group

The number of help given was not significantly influenced by the number of help received (main effect: $\beta \pm SE$: 0.11 ± 0.09, $P$ = 0.23, IRR (95% CI): 1.12 (0.92–1.34) in the 2013 distantly-related group. The number of help given was not significantly influenced by the pairwise genetic relatedness (main effect: $\beta \pm SE$: 17.32 ± 11.12, $P$ = 0.12, IRR (95% CI): 3.32e+7 (0.03–4.09e+17)). Absolute rank difference ($\beta \pm SE$: 0.05 ± 0.13, $P$ = 0.68, IRR (95% CI): 1.06 (0.82–1.40)), similarity in offspring sex ($\beta \pm SE$: 0.41 ± 0.42, $P$ = 0.33, IRR (95% CI): 1.51 (0.64–3.52)), and offspring birth mass difference ($\beta \pm SE$: -0.25 ± 0.19, $P$ = 0.18, IRR (95% CI): 0.78 (0.52–1.11)) did not significantly influence the number of help given. The intercept was not significant ($\beta \pm SE$: 0.77 ± 0.61, $P$ = 0.21). The random intercept effect for the first mother in a

dyad explained 1.00e-14 (SD = 1.00e-7) of the variance, and the random intercept effect for the other mother in a dyad explained 9.14e-16 (SD = 3.02e-8) of the variance. The model's conditional $R^2$ was equal to 0.25. There were 4 dyads with a tendency to reciprocate, i.e. RAFI $\geq$ 0.50.

The interaction between RAFI and pairwise genetic relatedness did not significantly affect the likelihood to allonurse ($\beta \pm SE$: -0.60 $\pm$ 30.52, $P$ = 0.98, OR (95% CI): 0.55 (5.76e-27– 5.23e25)) in the 2013 distantly-related group. The conditional main effect for RAFI when pairwise genetic relatedness was equal to 0 did not significantly influence the likelihood to allonurse ($\beta \pm SE$: 0.46 $\pm$ 1.40, $P$ = 0.74, OR (95% CI): 1.58 (0.10–24.63)). The conditional main effect for the pairwise genetic relatedness when RAFI was equal to 0 did not significantly influence the number of help given ($\beta \pm SE$: 0.01 $\pm$ 13.24, $P$ = 1.00, OR (95% CI): 1.01 (5.42e-12– 1.88e11)). The likelihood to allonurse was greater for male offspring than for female offspring (M vs F: $\beta \pm SE$: 1.44 $\pm$ 0.46, $P$ = 0.002, OR (95% CI): 4.22 (1.71–10.40)). Absolute rank difference ($\beta \pm SE$: 0.10 $\pm$ 0.23, $P$ = 0.66, OR (95% CI): 1.11 (0.70–1.73)), and offspring birth mass difference ($\beta \pm SE$: -0.12 $\pm$ 0.23, $P$ = 0.59, OR (95% CI): 0.89 (0.57–1.39)) did not significantly influence the likelihood to allonurse. The intercept was not significant ($\beta \pm SE$: -1.36 $\pm$ 0.98, $P$ = 0.17). The random intercept effect for the allonursing mother in a dyad explained 0.00 (SD = 0.00) of the variance, and the random intercept effect for the mother of the offspring being allonursed in a dyad explained 0.00 (SD = 0.00) of the variance. The model's conditional $R^2$ was equal to 0.14.

The likelihood of allonursing was not significantly influenced by RAFI (main effect: $\beta \pm SE$: 0.48 $\pm$ 0.83, $P$ = 0.56, OR (95% CI): 1.62 (0.32–8.22), Fig 2A) in the 2013 distantly-related group. The likelihood of allonursing was not significantly affected by the pairwise genetic relatedness (main effect: $\beta \pm SE$: -0.13 $\pm$ 11.28, $P$ = 0.99, OR (95% CI): 0.88 (2.20e-10–3.51e9), Fig 2B). The likelihood to allonurse was greater for male offspring than for female offspring (M vs F: $\beta \pm SE$: 1.44 $\pm$ 0.46, $P$ = 0.002, OR (95% CI): 4.22 (1.71–10.40)). Absolute rank difference ($\beta \pm SE$: 0.10 $\pm$ 0.23, $P$ = 0.66, OR (95% CI): 1.11 (0.70–1.73)) and offspring birth mass difference z-scored ($\beta \pm SE$: -0.12 $\pm$ 0.22, $P$ = 0.59, OR (95% CI): 0.89 (0.58–1.37)) did not significantly influence the likelihood of allonursing. The intercept was not significant ($\beta \pm SE$: -1.36 $\pm$ 0.95, $P$ = 0.15). The random intercept effect for the allonursing mother in a dyad explained 0.00 (SD = 0.00) of the variance, and the random intercept effect for the mother of the offspring being allonursed in a dyad explained 0.00 (SD = 0.00) of the variance. The model's conditional $R^2$ was equal to 0.14.

RAFI was not significantly influenced by pairwise genetic relatedness (main effect: $\beta \pm SE$: 6.58 $\pm$ 47.82, $P$ = 0.89, IRR (95% CI): 719.19 (1.41e-38–3.67e+43)) in the 2013 distantly-related group. The absolute difference in the age of mothers ($\beta \pm SE$: 1.13 $\pm$ 1.02, $P$ = 0.27, IRR (95% CI): 3.10 (0.42–22.89)) did not significantly influence RAFI. The intercept was marginally significant ($\beta \pm SE$: -12.43 $\pm$ 6.64, $P$ = 0.06). The random intercept effect for the first mother in a dyad explained 12.41 (SD = 3.52) of the variance, and the random intercept effect for the other mother in a dyad explained 126.31 (SD = 11.24) of the variance. The model's conditional $R^2$ was equal to 1.00.

## Meta-analyses for the direct reciprocity decision rule

For each additional help received by reindeer mothers, the rate of help given increased by 1.05 (fixed effects model for the incidence rate ratio: pooled effect size = 1.05 (95% CI: 1.01–1.08), p < 0.001, N = 3 effect sizes), which supports the direct reciprocity decision rule for the amount of help given and received. All effect sizes shared the same true effect size (Q = 2.00, df = 2, p = 0.37; $tau^2$ = 0.0001 (95% CI: 0.0000–0.2460); $I^2$ = 0.1% (95% CI: 0.0%– 89.6%);

H = 1.00 (95% CI: 1.00–3.10)). The effect size for the 2012 group accounted for 89.7% of the weight for the pooled effect size for the incidence rate ratio, whereas the 2013 closely-related and distantly-related groups accounted for 7.3% and 3.0%, respectively, of the weight for the pooled effect size. The odds of allonursing bout increased as RAFI increased (fixed effects model for OR: pooled effect size = 1.81 (95% CI: 1.38–2.39), p < 0.001, N = 3 effect sizes), which supports the direct reciprocity decision rule for the likelihood of a successful allonursing bout. All effect sizes shared the same true effect size (Q = 5.32, df = 2, p = 0.07; $tau^2$ = 0.27 (95% CI: 0.00–12.36); $I^2$ = 62.4% (95% CI: 0.0%– 89.3%); H = 1.63 (95% CI: 1.00–3.05)). The effect size for the 2012 group accounted for 87.0% of the weight for the pooled effect size for the OR, whereas the 2013 closely-related and distantly-related groups accounted for 10.1% and 2.8%, respectively, of the weight for the pooled effect size.

## Generalized reciprocity decision rule and the comparison between generalized reciprocity and direct reciprocity

The overall number of help given increased as the overall number of help received increased ($\beta \pm SE$: 0.004 ± 0.002, P = 0.033, IRR (95% CI): 1.0038 (1.0004–1.0077)). The intercept was significant ($\beta \pm SE$: -0.78 ± 0.27, P < 0.001). The random intercept effect for the mother explained 0.29 (SD = 0.54) of the variance, and the random intercept effect for the year explained 0.12 (SD = 0.34) of the variance. The model's conditional $R^2$ was equal to 0.29. The latency to give help after receiving help was shorter for generalized reciprocity than direct reciprocity by 18.31 days (mean ± SE latency for direct reciprocity: 19.42 ± 0.45 days (range: 1.00–64.09 days); mean ± SE latency for generalized reciprocity: 1.11 ± 0.04 days (range: 4.63e-5–16.31 days); model $\beta \pm SE$: -3.68 ± 0.07, P < 0.001, Fig 3). As the age of the mother increased, the latency to give help after receiving help decreased ($\beta \pm SE$: -0.17 ± 0.07, P = 0.017). Sex of the offspring (M vs F: $\beta \pm SE$: 0.09 ± 0.14, P = 0.51) and pairwise genetic relatedness ($\beta \pm SE$:

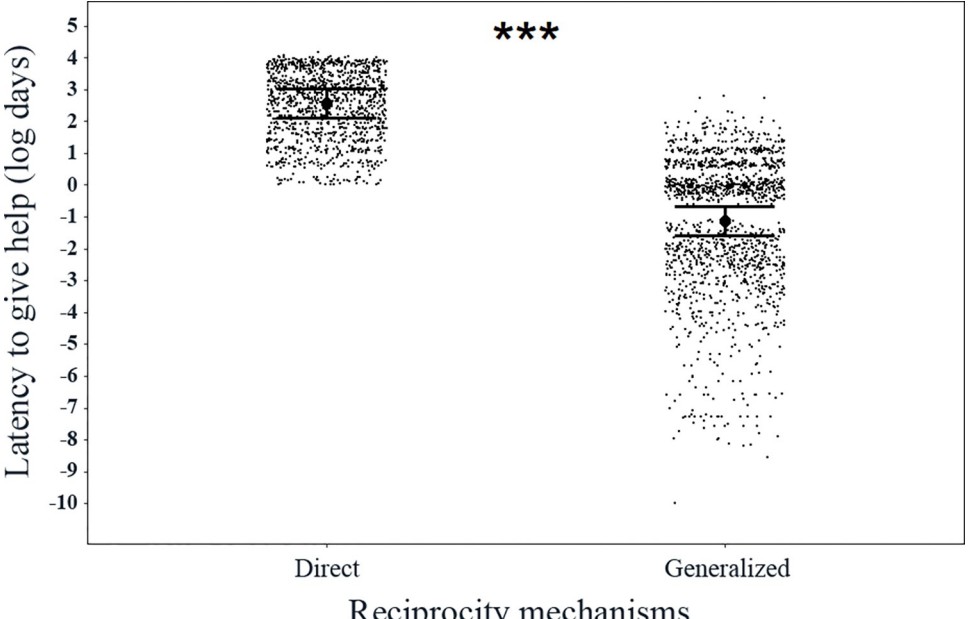

**Fig 3. The latency to give help after receiving help according to generalized reciprocity and direct reciprocity.** The latencies are log-transformed days. The central dots and whiskers represent the predicted values and the 95% confidence intervals. The black dots represent the raw data of food donations. ***P < 0.001.

0.13 ± 0.62, $P$ = 0.83) did not significantly influence the latency to give help after receiving help. The intercept was significant ($\beta \pm SE$: 2.52 ± 0.25, $P$ = 0.01). The random intercept effect for the mothers explained 0.08 (SD = 0.29) of the variance, and the random intercept effect for the groups explained 0.13 (SD = 0.36) of the variance. The model's conditional $R^2$ was equal to 0.56.

## Discussion

The positive association between help received and help given in pairs of reindeer mothers provided correlational evidence supporting the prediction for the direct reciprocity decision rule (prediction 1: main effect of help received on help given) in the 2012 group but not in both groups in 2013. These results were further supported by the positive association between the dyadic index of reciprocity, RAFI, and likelihood of allonursing, since the odds of a successful allonursing attempt were greater than the odds of an unsuccessful allonursing attempt as RAFI increased in the 2012 group but not in the 2013 groups (prediction 2: main effect of RAFI on the likelihood of allonursing). The results in the 2012 group provided further support for the partial evidence suggesting that reindeer mothers reciprocated allonursing at the group level, i.e. across bouts and dyads using matrix correlations [106]. A positive association between help given and help received within dyads is a commonly accepted result for allogrooming by direct reciprocity in primates [62,142,165] and in Norway rats [166], for food donations by direct reciprocity in Norway rats [61] and for food sharing by direct reciprocity in vampire bats [57]. The results suggest that reindeer mothers cooperate with social partners according to the evolved direct reciprocity decision rule (help someone who previously helped you) in the 2012 group. The internal meta-analysis results further supported the direct reciprocity decision rule among the three groups of reindeer mothers in 2012 and 2013, since 1) the rate of help given increased as help received increased, and 2) the odds of allonursing increased as RAFI increased among the 3 groups. The weight of the 2012 group for the pooled effect sizes was ≥ 87.0, which explains the significant internal meta-analysis results among the three groups. The direct reciprocity decision rule is an evolved decision rule of cooperation [20,21] that explains cooperation in various vertebrate taxa (mammals [28,29,56–62,64], birds [20,21] and fishes [54,55]) and microorganisms [66,67].

The question of "why was there no correlational evidence of reindeer mothers in both 2013 groups cooperating with social partners according to the direct reciprocity decision rule?" is unlikely to be explained by the cooperation mechanism of i) by-product mutualism [25,31–34], which cannot be cheated, since allonursing can be cheated by offspring stealing milk [7,108,167], and ii) enforcement [40,41], since neither the mothers nor the offspring coerced the other to transfer milk from the mother to the offspring. Allonursing can be a product of offspring stealing milk [6,7], and reindeer offspring do steal milk [108]. Thus, milk parasitism as a form of cooperation by deception plays a role in this communal breeding social system. Nonetheless, additional observation days were very likely required to assess the direct reciprocity decision rule in 2013. The difference between years is the time period of observation, i.e. 65 observation days in 2012 versus 25 observation days in 2013. The latency to give help after receiving help according to the direct reciprocity decision rule for allonursing reindeer mothers is on average 19.42 ± 0.45 days, yet the observation period spans 25 observation days in 2013. Returning allonursing help received was not observed to occur simultaneously, within a few minutes and within a day because i) lactation is the most energetically costly aspect of mammalian biology [8,9], and ii) reindeer have smaller udders than most ungulates [121].

Relatedness did not positively associate with help given in the 3 groups, which provided correlational evidence that did not support the prediction for the kin discrimination decision

rule (prediction 3: main effect of relatedness on help given). Two additional results further validated that reindeer mothers did not help each other by allonursing according to the kin discrimination decision rule. Relatedness did not positively associate with the likelihood of allonursing in the 3 groups, which provided correlational evidence that did not support the second prediction for the kin discrimination decision rule (prediction 4: main effect of relatedness on the likelihood of allonursing). The neutral associations between relatedness and RAFI in the three groups did not support the third prediction for the kin discrimination decision rule (prediction 5: main effect of relatedness on RAFI). These results suggest that reindeer mothers did not cooperate according to the kin discrimination decision rule, i.e. preferentially help kin. There are also neutral associations between relatedness and cooperation in correlational studies of eusocial and cooperative breeding social systems [88–94]. A negative association between relatedness and cooperation is supported by i) correlational studies of eusocial and cooperative breeding social systems [85–87], ii) a comparative analysis of eusocial hymenoptera [78], experimental manipulations of relatedness [29,87,97,98], and a theoretical model [99]. A larger range of relatedness is unlikely to positively associate with help given, the likelihood of allonursing and RAFI, since reindeer mothers rarely associate with related individuals during the calving season and during the rest of the year [135,168], except for mothers and their yearling adult daughters during the mating season. Thus, indirect fitness benefits are unlikely to be gained by cooperation in communally breeding reindeer.

We found no evidence to support the prediction for the effect of help received on help given depending on relatedness (prediction 6). Furthermore, the effect of RAFI on the likelihood of allonursing did not depend on relatedness (prediction 7). These results suggest that reindeer mothers did not cooperate with social partners according to an interaction between the direct reciprocity and kin discrimination decision rules. Cooperation in reindeer is better explained by the direct reciprocity decision rule than the kin discrimination decision rule. Direct reciprocity was relatively more important than relatedness in vampire bats [57] and in non-human, female primates [102]. Male Norway rats gave more help to unrelated social partners than to related social partners, and they gave more help to previously helpful social partners than previously unhelpful partners [29]. There was no evidence for a kinship bias i) for grooming in captive-born vampire bats forming new relationships, ii) for grooming rates in new and symmetrical relationships, and iii) for the emergence of reciprocal food sharing [58]. An interaction between reciprocity and relatedness has yet to be reported in a study of allonursing. A meta-analysis found that the incidence of allonursing was not associated with relatedness in groups where females associate with kin [15]. Several studies reported that females that allonurse associate with close kin allonurse the offspring of close kin [167,169–172], however only a few studies reported that allonursing contributions varied with relatedness [167,171,172]. Female dwarf mongooses breed communally with close relatives, and pregnant dwarf mongoose subordinates and spontaneous lactators allonursed close relatives [171]. Lion prides consist of closely related females, and the proportion of allonursing by lion mothers increased as the probability that all females in a crèche were first order relatives [167]. Within-group female relatedness in meerkats is high (0.41 ± 0.17), and the proportion of pregnant and recently pregnant females allonursing increased as the relatedness to the litter mother increased [172].

Receiving help from anyone increased help given to anyone, since there was a positive association between the overall number of help given and the overall number of help received by reindeer mothers increased. This result provided correlational evidence supporting the prediction for the generalized reciprocity decision rule (prediction 8), however the effect size was small. The latency to give help after receiving help according to the generalized reciprocity decision rule (help anyone after receiving help) is on average 1.11 ± 0.04 days. The latency to

give help after receiving help was shorter by 18.31 days according to the generalized reciprocity decision rule than according to direct reciprocity decision rule, which supported prediction 9. Older mothers returned received help sooner than younger mothers, and we suggest that older mothers may be more experienced in reciprocal interactions than younger mothers. The results suggest that reindeer mothers helped each other according to the generalized reciprocity decision rule, and they are the first evidence suggestive of the generalized reciprocity decision rule for allonursing and, more generally, alloparental care. The generalized reciprocity decision rule has been assessed in the greater spear-nosed bats (*Phyllostomus hastatus*) [173]. In the greater spear-nosed bats, alloparental care is not explained by kinship and mistaken identify, and it is unlikely to be explained by direct reciprocity and generalized reciprocity [173]. An interplay between cooperative foraging and group membership may explain alloparental care in greater spear-nosed bats [173]. A future experimental design is needed to assess if reindeer mothers allonurse according to generalized reciprocity and direct reciprocity decision rules or only according to one of these two decision rules. Both the direct reciprocity [28,35,36] and the generalized reciprocity [48,49,75,110,112] decision rules can lead to the evolution of cooperation.

Generalized reciprocity is cognitively less demanding than direct reciprocity, and generalized reciprocity does not require individual recognition and information about whether specific individuals previously helped them [21]. Humans help according to both direct reciprocity [56] and generalized reciprocity [68,69,174]. An experimental study found that female Norway rats, *Rattus norvegicus*, help more according to direct reciprocity than according to generalized reciprocity [70], however a meta-analysis found no apparent difference in the help given by female Norway rats to partners between the direct and generalized reciprocity decision rules [64]. Male Norway rats help according to direct reciprocity but not generalized reciprocity [64,73]. Swiss military dogs, *Canis lupus familiaris*, appear to help each other according to direct reciprocity, however the evidence supports that direct reciprocity is a byproduct of generalized reciprocity [71]. Long-tailed macaques help according to direct reciprocity and indirect reciprocity, i.e. help someone who is helpful (if A helped B, C helps A) but not according to generalized reciprocity, however spatial proximity may have affected the assessment of the generalized reciprocity decision rule [74]. Capuchin monkeys help according to generalized reciprocity [72]. Vampire bats help according to direct reciprocity but not generalized reciprocity [57]. The generalized reciprocity decision rule may be widespread and should be investigated in both correlational and experimental studies, yet there are few empirical studies of the generalized reciprocity decision rule in animals [60,64,68,70–72,74,175,176].

Reindeer offspring suckled 3 times/hr [126] or 1.7 times/hr [128] during the first week, and the frequency of suckling decreased throughout the lactation period. Researchers have reported allonursing in reindeer [122,126,128]. Espmark [126] recorded 85 allonursing bouts, where 14 of the 15 mothers allonursed, and all offspring solicitated to be allonursed. Six out of six mothers allonursed (290 allonursing bouts observed), and each of their offspring were allonursed [122]. The hypothesized causes of allonursing were not tested in these studies. Allonursing occurs more often in larger groups [6]. Allonursing incidence is likely high in wild reindeer and caribou, since they form very large herds. Sámi reindeer herders observe allonursing in large herds of semi-domesticated reindeer (personal communication with Sámi reindeer herders at the Sámi Education Institute in Inari and Kaamanen, Finland). Cooperation in reindeer may include babysitting, when a mother rests with the offspring while the other mothers forage, however we seldom observed this behaviour.

Reindeer are neither diurnal nor nocturnal, and their activity is not limited by daylight during the calving season. In the Arctic, there is daylight 24 h/day during the calving season. Nursing and allonursing in reindeer occur over 24 h/day based on preliminary observations.

Twenty-four hour monitoring of the reindeer mothers would increase the number of observed allonursing bouts and rejections, and the latency to return help received according to the direct reciprocity may be detected at a much shorter period than an average of 19.42 observation days. Such monitoring may also increase the effect size for the association between help received and help given according to the generalized reciprocity decision rules. Another limitation of this study is that we did not study the entire research population during the calving season. Future research of allonursing in reindeer should assess this behaviour in the context of the species' social organization and structure to identify additional variables that influence allonursing in reindeer.

Cooperation, like other behaviours, is an emergent property of evolved decision rules [19–21,24,28,35,47–50]. The study of evolved decision rules of cooperation and their interactions such as the kin discrimination, direct reciprocity and generalized reciprocity decision rules in the context of allonursing may further our understanding of cooperation in communal breeding social systems. These decision rules may be more common than previously believed. The generalized reciprocity decision rule has been ignored as an evolved decision rule in allonursing and has been seldom tested in the study of alloparental care. The alloparental care literature has rarely assessed if individuals help partners according to the direct reciprocity decision rule. The study of the interactions between these decision rules has also been ignored in the allonursing literature. Furthermore, there is yet no evidence to support that two females achieve a higher fitness when allonursing reciprocally than when they do not [7].

The latency to give help after receiving help extended to several days, i.e. on average 19.42 days, by direct reciprocity, however a long time delay hints to the possibility that reindeer may be capable of individual recognition, the ability to remember whether individuals previously helped them and the outcomes of previous encounters. Researchers believed that only humans met the requirements for direct reciprocity [52,53,177]. It was commonly believed that insects could not recognize other individuals, yet it occurs in insects [178–180]. Direct reciprocity in female Norway rats is mainly based on the outcome of the most recent encounter with a specific partner, independent of the last interaction preceding the test, as shown in a series of experience phases with different partners with the delay between help received and help given lasting up to 4 days [61,181]. These results highlight that Norway rats meet the required cognitive demands of direct reciprocity [61], and Norway rats apply the direct reciprocity decision rule rather than copying by imitation [63]. Future cognitive studies with reindeer as a model system should assess if reindeer meet the required cognitive demands of direct reciprocity, generalized reciprocity and kin discrimination.

## Conclusion

There was a positive association between the number of allonursing bouts given and the number of allonursing bouts received in 2012 group but not in both 2013 groups. There was also a positive association between the likelihood to allonurse and the extent to which pairs of mothers were reciprocal in the 2012 group but not in both 2013 groups: pairs of mothers with greater reciprocal index values were more likely to allonurse. The internal meta-analysis results and the correlational evidence suggested that semi-natural reindeer mothers allonurse according to the direct reciprocity rule (help someone who previously helped you). Correlation evidence for both the number of allonursing bouts given and the likelihood of allonursing suggested that reindeer mothers did not allonurse according to i) the kin discrimination decision rule, and ii) an interaction between the direct reciprocity and kin discrimination (preferentially help close kin) decision rules. We reported the first correlational evidence suggesting that semi-natural reindeer mothers allonurse according to the generalized reciprocity decision

rule (help anyone if helped by someone), and, more generally, the first evidence for the generalized reciprocity decision rule for alloparental care in non-human animals. The generalized reciprocity decision rule may be widespread and should be investigated in correlational and experimental studies. The latency to give after receiving help was shorter according to the generalized reciprocity decision rule than according to the direct reciprocity decision rule. These findings suggest that reindeer mothers cooperate by allonursing according to the direct reciprocity and generalized reciprocity decision rules.

## Supporting information

**S1 Data. The data for the number of help given in the 2012 group.**
(CSV)

**S2 Data. The data for the number of help given in the 2013 closely-related group.**
(CSV)

**S3 Data. The data for the number of help given in the 2013 distantly-related group.**
(CSV)

**S4 Data. The data for the likelihood to allonurse in the 2012 group.**
(CSV)

**S5 Data. The data for the likelihood to allonurse in the 2013 closely-related group.**
(CSV)

**S6 Data. The data for the likelihood to allonurse in the 2013 distantly-related group.**
(CSV)

**S7 Data. The data for the overall number of help given to assess the generalized reciprocity decision rule.**
(CSV)

**S8 Data. The data for the latency to give help after receiving help to compare the direct reciprocity and generalized reciprocity decision rules (.RData).** The data is named "Data8. RData". The.RData file can be opened in R with the function: load("Data8.RData"). If you have a problem downloading the file, please contact the corresponding author.
(RDATA)

**S9 Data. The data for the age of offspring in the three groups.**
(CSV)

**S1 File. The script for the analyses is named "Rscript.R".** The script can be read in R or Rstudio.
(R)

## Acknowledgments

We thank LUKE (Natural Resources Institute Finland), Mika Tervonen, Ilmo Pehkonen, Erkki Paltto, Heikki Törmänen, Jukka Siitari, and the Finnish Reindeer Herding Association for their help and advice. We thank Prof. Michael Taborsky and two anonymous reviewers for their comments.

## Author Contributions

**Conceptualization:** Sacha C. Engelhardt, Robert B. Weladji, Øystein Holand, Knut H. Røed, Mauri Nieminen.

**Data curation:** Sacha C. Engelhardt.

**Formal analysis:** Sacha C. Engelhardt.

**Funding acquisition:** Robert B. Weladji, Øystein Holand, Knut H. Røed.

**Investigation:** Sacha C. Engelhardt.

**Methodology:** Sacha C. Engelhardt, Robert B. Weladji, Øystein Holand, Knut H. Røed, Mauri Nieminen.

**Resources:** Knut H. Røed, Mauri Nieminen.

**Supervision:** Robert B. Weladji, Øystein Holand, Knut H. Røed.

**Validation:** Sacha C. Engelhardt.

**Visualization:** Sacha C. Engelhardt.

**Writing – original draft:** Sacha C. Engelhardt, Robert B. Weladji, Øystein Holand, Knut H. Røed, Mauri Nieminen.

**Writing – review & editing:** Sacha C. Engelhardt, Robert B. Weladji, Øystein Holand, Knut H. Røed, Mauri Nieminen.

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
