## [Decision Letter · Decision Letter 0]

1 Feb 2023

PONE-D-22-28248Evidence suggesting that reindeer mothers allonurse according to the direct reciprocity and generalized reciprocity decision rulesPLOS ONE

Dear Dr. Engelhardt,

Thank you for submitting your manuscript to PLOS ONE. After careful consideration, we feel that it has merit but does not fully meet PLOS ONE’s publication criteria as it currently stands. Therefore, we invite you to submit a revised version of the manuscript that addresses the points raised during the review process.

We look forward to receiving your revised manuscript.

Kind regards,

Anindita Bhadra, PhD

Academic Editor

PLOS ONE

and https://journals.plos.org/plosone/s/file?id=ba62/PLOSOne_formatting_sample_title_authors_affiliations.pdf.

2. Thank you for including your ethics statement: " We designed this study in accordance with the Animal Ethics and Care Certificate of Concordia University (AREC-2010-WELA) and the Finnish National Advisory Board on Research Ethics.". To comply with PLOS ONE submissions requirements, please provide the following information in the Methods section of the manuscript and in the “Ethics Statement” field of the submission form (via “Edit Submission”): * Please indicate whether an animal research ethics committee prospectively approved this research or granted a formal waiver of ethics approval.* Please enter the name of your Institutional Animal Care and Use Committee (IACUC) or other relevant ethics board. Also include an approval number if one was obtained. * If anesthesia, euthanasia, or any kind of animal sacrifice is part of the study, please include briefly in your statement which substances and/or methods were applied. For additional information about PLOS ONE submissions requirements for ethics oversight of animal work, please refer to http://journals.plos.org/plosone/s/submission-guidelines#loc-animal-research Once you have amended this/these statement(s) in the Methods section of the manuscript, please add the same text to the “Ethics Statement” field of the submission form (via “Edit Submission”).

“This project was supported by a doctoral bursary from the Fonds de recherche du Québec, Nature et technologies, a Northern Scientific Training Program award, and a QCBS Excellence Award granted to SCE. This project was also funded by RB's Natural Sciences and Engineering Research Council of Canada (https://www.nserc-crsng.gc.ca/index_eng.asp) Discovery Grant number 327505. The contributions of KR and ØH were funded by Reindeer Husbandry in a Globalizing North (ReiGN), which is a Nordforsk-funded (https://www.nordforsk.org/nordic-centre-excellence) “Nordic Centre of Excellence” (project number 76915).”

4. We noted in your submission details that a portion of your manuscript may have been presented or published elsewhere. [No. The data have not been published elsewhere, however we combined data from 2 years of data collection. We published articles using parts of the data used in this manuscript. Some of data from Engelhardt et al. (2015) in Ethology is included in the data of the current manuscript. We describe in the methods that we revise the partial evidence suggesting that reindeer mothers allonurse according to the direct reciprocity decision rule (Engelhardt et al. 2015). The current manuscript now shows evidence suggesting that reindeer mothers do allonurse according to the direct reciprocity and generalized reciprocity decision rule. No results or figures were previously published or in pending manuscripts.

Please clarify whether this [conference proceeding or publication] was peer-reviewed and formally published. If this work was previously peer-reviewed and published, in the cover letter please provide the reason that this work does not constitute dual publication and should be included in the current manuscript.”

Additional Editor Comments:

This manuscript needs extensive revision. I am sorry that the decision took long, due to the difficulty of finding suitable reviewers. I agree with Reviewer 2 regarding the issue of data collection, and I suggest that the manuscript is revised keeping the detailed comments in mind. Analyzing the data for the two years separately is a good suggestion. I look forward to receiving the revised manuscript in the near future.

Reviewers' comments:

Reviewer's Responses to Questions

**Comments to the Author**

1. Is the manuscript technically sound, and do the data support the conclusions?

Reviewer #1: Yes

Reviewer #2: No

2. Has the statistical analysis been performed appropriately and rigorously? 

Reviewer #1: Yes

Reviewer #2: No

3. Have the authors made all data underlying the findings in their manuscript fully available?

Reviewer #1: Yes

Reviewer #2: No

4. Is the manuscript presented in an intelligible fashion and written in standard English?

Reviewer #1: Yes

Reviewer #2: Yes

5. Review Comments to the Author

Reviewer #1: This article represents the existence and nature of allosuckling behaviour in reindeer which could add important perspective to the allocare behaviour in mammals. The authors have used various statistical tests and models to justify their findings. Please see the review report for some minor issues which should be considered by the authors prior to the final publication of this manuscript.

Reviewer #2: The review comments have been included as an attachment, which includes an overview of the comments as well as detailed comments for all the sections of the manuscript. Please refer to the attached PDF for a complete review.

6. PLOS authors have the option to publish the peer review history of their article (what does this mean?). If published, this will include your full peer review and any attached files.

Reviewer #1: No

Reviewer #2: No

---

## [Author Response · Author response to Decision Letter 0]

1 Apr 2023

We attached our response to reviewers document with figures and tables and Rscript outputs. We assume that we do not need to copy and paste all of this here.

---

## [Decision Letter · Decision Letter 1]

22 Jun 2023

PONE-D-22-28248R1Evidence suggesting that reindeer mothers allonurse according to the direct reciprocity and generalized reciprocity decision rulesPLOS ONE

Dear Dr. Engelhardt,

Thank you for submitting your manuscript to PLOS ONE. After careful consideration, we feel that it has merit but does not fully meet PLOS ONE’s publication criteria as it currently stands. Therefore, we invite you to submit a revised version of the manuscript that addresses the points raised during the review process.

We look forward to receiving your revised manuscript.

Kind regards,

Anindita Bhadra, PhD

Academic Editor

PLOS ONE

Journal Requirements:

Additional Editor Comments:

I am sorry that the reviewing process took a long time, primarily due to the difficulty of finding reviewers. Please revise your manuscript, especially keeping the suggestions from Reviewer 3 in mind. I look forward to receiving the revised manuscript soon.

Reviewers' comments:

Reviewer's Responses to Questions

**Comments to the Author**

1. If the authors have adequately addressed your comments raised in a previous round of review and you feel that this manuscript is now acceptable for publication, you may indicate that here to bypass the “Comments to the Author” section, enter your conflict of interest statement in the “Confidential to Editor” section, and submit your "Accept" recommendation.

Reviewer #1: All comments have been addressed

Reviewer #3: (No Response)

2. Is the manuscript technically sound, and do the data support the conclusions?

Reviewer #1: Yes

Reviewer #3: Yes

3. Has the statistical analysis been performed appropriately and rigorously? 

Reviewer #1: Yes

Reviewer #3: Yes

4. Have the authors made all data underlying the findings in their manuscript fully available?

Reviewer #1: Yes

Reviewer #3: (No Response)

5. Is the manuscript presented in an intelligible fashion and written in standard English?

Reviewer #1: Yes

Reviewer #3: Yes

6. Review Comments to the Author

Reviewer #1: Line 39-41: Statements are unclear until we read the lines 42-43. Please revise this section before final publication.

This article represents the existence and nature of allosuckling behaviour in reindeer which could add important perspective to the allocare behaviour in mammals. The authors have used now justified their findings and addressed all the revisions required for the publication.

Reviewer #3: This paper reveals that reindeer mothers allonurse each others’ offspring by applying direct and generalised reciprocity, but not by kin discrimination. These are intriguing and important results challenging common belief that relatedness effects exceed effects of received service on altruism. In other words, reciprocity seems to explain allonursing in reindeer mothers, whereas relatedness does not. In its own right, the observation that generalised reciprocity can partly explain the propensity to allonurse other mothers’ offspring is striking. Other interesting results of this study include the observation that applying the generalised reciprocity rule is associated with shorter latencies between received and given help than applying the direct reciprocity rule, which corroborates theoretical predictions of a faster decline of effects from received help in the anonymous than in the specific partner situations; and older mothers were responding quicker to received help than younger mothers, which is a result that is not discussed, unfortunately. Overall, this is an important study that deserves being published.

Nevertheless, there is scope for improvement. The Results section is partly hard to read because of putative redundancy and overloading. I strongly recommend to structure it more clearly (e.g. deal with the different datasets from 2012 and 2013 (the latter split by relatedness structure differences) in separate blocks that are clearly marked with respective sub-headings). Now the reader must always search for specific information about which dataset is actually just referred to. I recommend to shift some of the results in an Appendix, so that the really important information can be grasped much more easily in the main text.

I provide a number of recommendations to improve the manuscript in my detailed comments below.

Detailed comments (by line numbers):

26-28 and 98: This reads as if these three decision rules are the only possibility to explain the evolution of cooperation, which – even if being of fundamental importance - I think is not really true (cf. Fig. 4.2 in ref. 20, which summarises the selection mechanisms underlying cooperation).

41: after “as the overall number of help received” the word “increased” should be added.

47: I propose to replace “feeding” by “caring for” (e.g. in cooperatively breeding fishes the young are cared for, but not fed by parents and alloparents).

63: “Allonursing is more common in polytocous” add “species”.

82-83: I am not sure that Hamilton 1964 (II) has outlined all three mechanisms.

86-87: I propose to replace “first” by “originally”.

95-96: An example is given by ref. 42.

101-102: “related kin” is a pleonasm.

104: “evolutionarily”

106: “use of environmental”

108-109: Perhaps “in previous social interactions” (plural) would be better.

115 “not supported in male Norway rats [55]”: Perhaps one should add here (“as opposed to females”).

117: I suggest to make a paragraph break after “serial reciprocity [62]”.

121: I would write “or neutral”.

125: I suggest to replace “differs” by “contrasts”.

126 “and their fitness pay-offs are correlated”: This is not entirely correct. Not the fitness pay-offs of “reciprocity and relatedness” are correlated, but those of the interacting agents.

129: I would say “in some empirical studies”.

130-131: In dogs as well (cf. ref. 53).

145: Perhaps “rejected their own offspring” would be clearer.

151: I agree, but it would be good to justify this need (e.g. because there is a number of theoretical models showing that it can create evolutionarily stable levels of cooperation, and because it is based on such a simple decision rule that virtually any animal species should be able to apply it).

165-166 “the likelihood to allonurse should increase the reciprocity index values of pairs of mothers increases”: this sentence is grammatically incorrect. Also, the “reciprocity index values” have not yet been explained.

170: Spello (reciporocity).

176: “subsequent to help given”.

179-180: Why? I would have predicted the opposite (because direct reciprocity should generate a higher propensity to provide help than generalised reciprocity; it is less prone to cheating).

458-459: “… did not significantly influence”

470-471 and 492-493: I am afraid I do not understand how these three expected numbers are derived given the factor of increase is only 1.05 or 1.10, respectively. Why does the number of help of the receiver increase LESS (2.48 against 2.61) for each help received if the multiplication factor is higher (1.1 vs. 1.05)? Also, I would write “1” instead of “one” when referring to actual results.

502-511: Perhaps for readability it would be good to start this long sentence with its main message that all these variables “did not significantly influence the number of help given in the 2013 closely-related group”.

515-516 and 527-528: “increased” seems redundant/ grammatically incorrect. Also, why is the potential dependency turned around here? Usually, you checked whether the help given is influenced by the help received, but here it seems opposite. If you look at this both ways, does this mean double testing the same data (even if from different ends)?

515-526 and 527-538: These two paragraphs seem to relate to the same dataset, “the 2013 distantly-related group”. What is the difference?

553-554, 582-583 and 608-609: I’m afraid this is tautological. RAFI is a measure of reciprocal allonursing, so a higher index value inadvertently means an “increased likelihood of allonursing”, doesn’t it?

554-555, 583-585 and 609-611: Without further specification this is redundant (pairwise genetic relatedness effects on the likelihood of allonursing were scrutinised already further above).

Fig. 2 seems redundant as explained by my last two comments.

573 and 659: “did not significantly influence”

595-597 “The conditional main effect for RAFI when pairwise genetic relatedness was equal to 0 significantly influenced the likelihood to allonurse”: How can this be justified by a P-value of 0.74?

640: “… as the offspring birth mass difference increased”

677-678 “A positive association between help given and help received within dyads is a commonly accepted result for allogrooming by direct reciprocity in primates”: This applies also to Norway rats (Stieger et al. 2017, Behavioral Ecology and Sociobiology 71: 182).

679 “… and for food sharing by direct reciprocity in vampire bats”: And Norway rats (ref. 45).

686: What is “a different evolved explanation for cooperation”?

688: “give” (not “given”).

720-721: The logic of correlated pay-offs is usually applied to the effects of cooperation on the partners involved; their pay-offs may be correlated by a reciprocal exchange of service in iterated interactions, or by sharing genes (cf. ref. 21). I am not aware of an expectation that “direct reciprocity and relatedness” should “have correlated pay-offs”.

730-731: There is some grammatical problem in this sentence; as such it does not make sense.

755: “… according to one”

755: I think ref. 32 should be mentioned here as well.

756: Ref. 30 would seem adequate here as well, and perhaps also Rankin & Taborsky 2009 (Evolution 63, 1913-1922) because it modelled the evolution of generalised reciprocity under most general conditions.

758: I suggest writing something like “… and information about whether specific individuals …”

765-766: The study in longtailed macaques was not well suited test for generalised reciprocity because spatial proximity was not controlled for (cf. p. 151 in ref. 20).

769-770: There were some more experimental studies, e.g. Schneeberger et al. 2012 (BMC Evolutionary Biology 2012, 12:41) and Gerber et al. 2020 (Proc. R. Soc. B 287: 20202327).

778: Can you cite evidence for the herders’ observations?

798: How can a “study” be “tested”?

801: “Roulin 2002” does not comply with the usual citation format.

803-805 “a long time delay does not exclude the possibility…”: I would rather suggest it “hints on this possibility”. Without this long-term memory, these results could not be easily explained.

939: This reference (47) needs to be corrected.

Michael Taborsky

(I always sign my reviews because I am in favour of transparency in science; cf. Ethology 113 (2007), 1–8)

7. PLOS authors have the option to publish the peer review history of their article (what does this mean?). If published, this will include your full peer review and any attached files.

Reviewer #1: No

Reviewer #3: **Yes: **Michael Taborsky

---

## [Author Response · Author response to Decision Letter 1]

19 Jul 2023

PONE-D-22-28248R1

Evidence suggesting that reindeer mothers allonurse according to the direct reciprocity and generalized reciprocity decision rules

PLOS ONE

Dear Dr. Engelhardt,

Thank you for submitting your manuscript to PLOS ONE. After careful consideration, we feel that it has merit but does not fully meet PLOS ONE’s publication criteria as it currently stands. Therefore, we invite you to submit a revised version of the manuscript that addresses the points raised during the review process.

We look forward to receiving your revised manuscript.

Kind regards,

Anindita Bhadra, PhD

Academic Editor

PLOS ONE

Journal Requirements:

Additional Editor Comments:

I am sorry that the reviewing process took a long time, primarily due to the difficulty of finding reviewers. Please revise your manuscript, especially keeping the suggestions from Reviewer 3 in mind. I look forward to receiving the revised manuscript soon.

R1_AE: We include the following updated statement in our cover letter: “The funders had no role in study design, data collection and analysis, decision to publish, or preparation of the manuscript. Please add the following to our Financial Disclosure statement: “The funders had no role in study design, data collection and analysis, decision to publish, or preparation of the manuscript.”” We stated that the funders had no role in study design, data collection and analysis, decision to publish, or preparation of the manuscript in the previous cover letter. We now request that this be added to our Financial Disclosure statement. 

We revised the manuscript according to your comments and the reviewers’ comments.

We reviewed our reference list to ensure it is complete and correct.

We copied and pasted our response R32_RE3 to Reviewer 3’s 32nd comment. We would be willing to move the likelihood of allonursing results and Figure 2 to the Appendix, if Reviewer 3 and/or the Academic Editor would prefer that. We left the results and Figure 2 in the main text for the reasons described in our response R32_RE3, which follow. The reciprocity index, RAFI, is based on successful allonursing bouts given to each other by both mothers in a dyad. However, this index does not account for the allonursing rejections by mothers. The number of help received and the number of help given by mothers accounts for successful allonursing bouts received and given by mothers without accounting for the allonursing rejections by mothers. As such, we assessed whether more reciprocal dyads of mothers may be more likely to allonurse, i.e. the odds of a successful allonursing attempt may be greater than the odds of an unsuccessful allonursing attempt for more reciprocal dyads. If the likelihood of allonursing is influenced by RAFI, then this suggests that the odds of a successful allonursing attempt are greater for more reciprocal dyads, i.e. higher RAFI values, than the odds of an unsuccessful allonursing attempt. If there were no significant association between RAFI and the likelihood of allonursing, the odds of a successful allonursing attempt would not differ from the odds of an unsuccessful allonursing attempt, and this could happen if offspring attempted and were rejected just as often as they were accepted, independent of the dyadic reciprocity index. Since the odds of a successful allonursing attempt are greater than the odds of an unsuccessful allonursing attempt as RAFI increases, we view this as an important result that further supports the positive association between the amount of help received and the amount of help given by mother, which supports that reindeer mothers help partners according to the direct reciprocity decision rule. We hope that we have convinced Reviewer 3 and the Academic Editor that this is not tautology but rather interesting and important results.

We wrote the following in the methods on lines 393-399: “The number of help received and the number of help given by mothers accounts for successful allonursing bouts received and given by mothers without accounting for the allonursing rejections by mothers, i.e. unsuccessful allonursing attempts, however the likelihood of allonursing model accounts for both the successful and unsuccessful allonursing attempts. We assessed whether more reciprocal dyads of mothers may be more likely to allonurse, i.e. the odds of a successful allonursing attempt may be greater than the odds of an unsuccessful allonursing attempt for more reciprocal dyads.”

We wrote the following in the discussion on lines 703-706: “These results were further supported by the positive association between the dyadic index of reciprocity, RAFI, and likelihood of allonursing, since the odds of a successful allonursing attempt were greater than the odds of an unsuccessful allonursing attempt as RAFI increased in the 2012 group but not in the 2013 groups (prediction 3: main effect of RAFI on the likelihood of allonursing).”

We would be willing to move the likelihood of allonursing results to the Appendix, if Reviewer 3 and/or the Academic Editor would prefer that.

Reviewers' comments:

Reviewer's Responses to Questions

Comments to the Author

1. If the authors have adequately addressed your comments raised in a previous round of review and you feel that this manuscript is now acceptable for publication, you may indicate that here to bypass the “Comments to the Author” section, enter your conflict of interest statement in the “Confidential to Editor” section, and submit your "Accept" recommendation.

Reviewer #1: All comments have been addressed

Reviewer #3: (No Response)

2. Is the manuscript technically sound, and do the data support the conclusions?

Reviewer #1: Yes

Reviewer #3: Yes

3. Has the statistical analysis been performed appropriately and rigorously?

Reviewer #1: Yes

Reviewer #3: Yes

4. Have the authors made all data underlying the findings in their manuscript fully available?

Reviewer #1: Yes

Reviewer #3: (No Response)

5. Is the manuscript presented in an intelligible fashion and written in standard English?

Reviewer #1: Yes

Reviewer #3: Yes

6. Review Comments to the Author

1): Reviewer #1: Line 39-41: Statements are unclear until we read the lines 42-43. Please revise this section before final publication.

R1_RE1: We revised this section on lines 39-45: “The number of help given i) increased as the number of help received increased in the 2012 group but not in both 2013 groups, ii) was not influenced by relatedness, and iii) was not influenced by an interaction between the number of help received and relatedness. The overall number of help given increased as the overall number of help received increased. The results did not support the prediction that reindeer mothers allonursed according to the kin discrimination decision rule. The results suggest that reindeer mothers may allonurse according to the direct reciprocity and generalized reciprocity decision rules.”

2): This article represents the existence and nature of allosuckling behaviour in reindeer which could add important perspective to the allocare behaviour in mammals. The authors have used now justified their findings and addressed all the revisions required for the publication.

R2_RE1: Thank you for this comment.

1) Reviewer #3: This paper reveals that reindeer mothers allonurse each others’ offspring by applying direct and generalised reciprocity, but not by kin discrimination. These are intriguing and important results challenging common belief that relatedness effects exceed effects of received service on altruism. In other words, reciprocity seems to explain allonursing in reindeer mothers, whereas relatedness does not. In its own right, the observation that generalised reciprocity can partly explain the propensity to allonurse other mothers’ offspring is striking. Other interesting results of this study include the observation that applying the generalised reciprocity rule is associated with shorter latencies between received and given help than applying the direct reciprocity rule, which corroborates theoretical predictions of a faster decline of effects from received help in the anonymous than in the specific partner situations; and older mothers were responding quicker to received help than younger mothers, which is a result that is not discussed, unfortunately. Overall, this is an important study that deserves being published.

R1_RE3: Thank you for this comment. 

We wrote the following in the discussion on lines 781-783: “Older mothers may attempt to foster more social interactions with potential partners to receive more help in the future by giving help sooner after receiving help than younger mothers.”

2) Nevertheless, there is scope for improvement. The Results section is partly hard to read because of putative redundancy and overloading. I strongly recommend to structure it more clearly (e.g. deal with the different datasets from 2012 and 2013 (the latter split by relatedness structure differences) in separate blocks that are clearly marked with respective sub-headings). Now the reader must always search for specific information about which dataset is actually just referred to. I recommend to shift some of the results in an Appendix, so that the really important information can be grasped much more easily in the main text.

R2_RE3: We added sub-headings as recommended for each of the 3 groups, so the important information can be grasped more easily in the main text: “Decision rules of direct reciprocity, kin discrimination and their interaction for the 2012 group”, “Decision rules of direct reciprocity, kin discrimination and their interaction for the 2013 closely-related group”, “Decision rules of direct reciprocity, kin discrimination and their interaction for the 2013 distantly-related group” on lines 477-478, 569-570 and 630-631. 

I provide a number of recommendations to improve the manuscript in my detailed comments below.

Detailed comments (by line numbers):

3) 26-28 and 98: This reads as if these three decision rules are the only possibility to explain the evolution of cooperation, which – even if being of fundamental importance - I think is not really true (cf. Fig. 4.2 in ref. 20, which summarises the selection mechanisms underlying cooperation).

R3_RE3: We added “by at least” to the following in the abstract on lines 28-31: “Cooperation can be explained by at least three evolved decision rules: 1) direct reciprocity, i.e. help someone who previously helped you, 2) kin discrimination, i.e. preferentially direct help to kin than to non-kin, and 3) generalized reciprocity, i.e. help anyone if helped by someone.” 

We agree with Reviewer 3 that multiple mechanism can underlying cooperation. We revised the introduction on lines 101-110: “Lehman and Keller [30] proposed a theoretical framework and classification of models for the evolution of cooperation. These models can be divided into four distinct categories [20]. Cooperation can arise from the net fitness benefits of i) by-product mutualism [25,31–34]. Cooperation can evolve from the net fitness benefits of altruism according to the correlated pay-offs of conditional returns due to ii) an above-random chance that help provided to a social partner will increase the likelihood of receiving help in return in the future, i.e. reciprocity [21,28,35,36]; or by the correlated pay-offs of shared genes iii) kin selection [19,24,37–39]. Cooperation can evolve by manipulation according to enforcement [40,41] or deception [42–46]. Cooperation is an emergent property of evolved decision rules [19–21,24,28,35,47–50]. Cooperation can be explained by at least three evolved, mechanistic decision rules …”

4) 41: after “as the overall number of help received” the word “increased” should be added.

R4_RE3: We added “increased” in the abstract on lines 41-42: “The overall number of help given increased as the overall number of help received increased.”

5) 47: I propose to replace “feeding” by “caring for” (e.g. in cooperatively breeding fishes the young are cared for, but not fed by parents and alloparents).

R5_RE3: We replaced “feeding” by “caring for” on lines 48-49: “Cooperative breeding can be defined as breeding females assisted in protecting and caring for their offspring by non-breeding helpers”

6) 63: “Allonursing is more common in polytocous” add “species”.

R6_RE3: We revised this sentence as follows on lines 65-67: “Allonursing is more common in polytocous species, i.e. giving birth to more than one offspring per parturition, than monotocous species, i.e. giving birth to one offspring per parturition [6,15].”

7) 82-83: I am not sure that Hamilton 1964 (II) has outlined all three mechanisms.

R7_RE3: We added Hamilton’s (1964) The Genetical Evolution of Social Behaviour. I and wrote the following on lines 83-85: “Kin selection is selection acting on the consequences of an individual’s behaviour on the survival and/or reproduction of its relatives [22,23], and its three mechanisms are kin discrimination, limited dispersal and greenbeards [19,24].” References 19 and 24 refer to Hamilton’s (1964a, 1964b) The Genetical Evolution of Social Behaviour. I and The Genetical Evolution of Social Behaviour. II.

In Hamilton’s (1964) The Genetical Evolution of Social Behaviour. II, section four “Discrimination in social situations” describes the general concept of kin discrimination “If he could learn to recognize those of his neighbours who really were close relatives and could devote his beneficial actions to them alone an advantage to inclusive fitness would at once appear.” This same section refers to “The selective advantage when a benefit comes to be given to sibs only instead of to sibs and half-sibs indifferently is more than four times the advantage when a benefit of the same magnitude is given to cousins only instead of to cousins and half-cousins indifferently. Nevertheless, if any correlate of relationship is very persistent, long continued weak selection could lead to the evolution of a discrimination based on it even in the range of distant relationships. One possible factor of this kind in species with viscous populations, and one whose persistence depends only on the viscosity and therefore may well be considerably older than the species in question, is familiarity of appearance. For in a viscous population the organisms of a particular neighbourhood, being relatives, must tend to look alike and an individual which used the restrained symbolic forms of aggressive behaviour only towards familiar-looking rivals would be effecting a discrimination advantageous to inclusive fitness.” There are additional examples in this section.

The greenbeard effect in the section four: “That genes could cause the perception of the presence of like genes in other individuals may sound improbable; at simplest we need to postulate something like a supergene affecting (a) some perceptible feature of the organism, (b) the perception of that feature, and (c) the social response consequent upon what was perceived.” Dawkins (1976) popularized the greenbeard.

Population viscocity (aka limited dispersal) is referred to in Hamilton’s (1964) The Genetical Evolution of Social Behaviour. I: “With many natural populations it must happen that an individual forms the centre of an actual local concentration of his relatives which is due to a general inability or disinclination of the organisms to move far from their places of birth. In such a population, which we may provisionally term “viscous”, the present form of selection may apply fairly accurately to genes which affect vagrancy. It follows from the statements of the last paragraph but one that over a range of different species we would expect to find giving traits commonest and most highly developed in the species with the most viscous populations whereas uninhibited competition should characterize species with the most freely mixing populations.” Hamilton also refers to viscous populations in The Genetical Evolution of Social Behaviour. I “The selective forces operating ‘on the post-reproductive life-span are doubtless generally weak; they will be strongest when the average relationship of neighbours is highest, which will be in the most viscous populations.”

8) 86-87: I propose to replace “first” by “originally”.

R8_RE3: We replaced “first” with “originally” on lines 87-90: “The indirect fitness benefits as a result of helping appear to be smaller than originally expected or may be underestimated, and the direct fitness benefits as a result of help appear to be greater than originally expected and may even be greater than the indirect fitness benefits [21,25,26].”

9) 95-96: An example is given by ref. 42.

R9_RE3: We included the reference and wrote the following on lines 97-100: “Reciprocal interactions can also occur between related and unrelated individuals, such that the correlated pay-offs for the interacting partners can interact with relatedness [21], e.g. in female Norway rats, Rattus norvegicus [29].”

10) 101-102: “related kin” is a pleonasm.

R10_RE3: We removed “related” and now reads as the following on lines 113-114: “… preferentially direct help to kin than to non-kin (if A and B are kin, but A and C are non-kin, A preferentially helps B)”

11) 104: “evolutionarily”

R11_RE3: We made the change on line 115-116: “These decision rules can generate evolutionarily stable cooperation [19–21,24,28,35,47–49].”

12) 106: “use of environmental”

R12_RE3: We made the change on line 116-118: “Kin discrimination requires that individuals can discriminate relatives from non-relatives and occurs by the use of environmental or genetic cues [51].”

13) 108-109: Perhaps “in previous social interactions” (plural) would be better.

R13_RE3: We made the change on lines 119-121: “Generalized reciprocity only requires the ability to remember if one received help or not in previous social interactions, without needing to remember and identify the partner(s) [20,21].”

14) 115 “not supported in male Norway rats [55]”: Perhaps one should add here (“as opposed to females”).

R14_RE3: We refer to both female and male Norway rats in the sentence on lines 125-128: “Generalized reciprocity was reported in humans [66,67], female Norway rats [60,68], dogs, Canis familiaris [69], capuchin monkeys, Sapajus apella [70], however generalized reciprocity was not supported in male Norway rats [71], longtailed macaques, Macaca fascicularis [72] and in vampire bats [57].” 

15) 117: I suggest to make a paragraph break after “serial reciprocity [62]”.

R15_RE3: We broke the paragraph into two paragraphs on lines 128-130: “Generalized reciprocity is also known as upstream tit-for-tat [47], upstream indirect reciprocity [73], upstream reciprocity [74], pay it forward [75–77], and serial reciprocity [78].” 

Most of the evidence for the importance of kin selection …”

16) 121: I would write “or neutral”.

R16_RE3: We made the change on lines 131-133: “For empirical studies in taxa with a high social complexity, i.e. eusocial and cooperative breeding social systems, the association between relatedness and cooperation was positive [79–82], negative [83–85] or neutral [86–92].”

17) 125: I suggest to replace “differs” by “contrasts”.

R17_RE3: We made the change on lines 136-138: “Several studies that experimentally manipulated relatedness found that relatedness reduced cooperation rather than increase it [29,85,95,96], which contrasts with kin selection theory.”

18) 126 “and their fitness pay-offs are correlated”: This is not entirely correct. Not the fitness pay-offs of “reciprocity and relatedness” are correlated, but those of the interacting agents.

R18_RE3: We revised the sentence and wrote the following on lines 138-139: “Cooperation can be explained by reciprocity and relatedness, and the fitness pay-offs of the interacting partners are correlated [20,21].”

19) 129: I would say “in some empirical studies”.

R19_RE3: We made the change on lines 141-142: “Cooperation was better explained by direct reciprocity than relatedness [29,57,100] and generalized reciprocity [72] in some empirical studies.”

20) 130-131: In dogs as well (cf. ref. 53).

R20_RE3: We wrote the following on lines 142-144: “Direct reciprocity and generalized reciprocity can co-exist, such as in humans [66], female Norway rats [60,68], and in dogs [69,101].”

21) 145: Perhaps “rejected their own offspring” would be clearer.

R21_RE3: We made the change on lines 157-158: “Reindeer mothers rejected offspring attempting to allosuckle more often than they rejected their own offspring attempting to suckle [106].”

22) 151: I agree, but it would be good to justify this need (e.g. because there is a number of theoretical models showing that it can create evolutionarily stable levels of cooperation, and because it is based on such a simple decision rule that virtually any animal species should be able to apply it).

R22_RE3: We wrote the following on lines 162-169: “There are few empirical studies of the generalized reciprocity decision rule in animals [60,66,68–70,72], and we need to further our knowledge of how widespread the generalized reciprocity decision rule is in animals. Most animal species should be able to apply the generalized reciprocity decision rule, since several theoretical models found that generalized reciprocity can generate evolutionarily stable levels of cooperation [21,47,111,112,48,49,73–75,108–110], and the decision rule of generalized reciprocity is less cognitively demanding, i.e. a simpler decision rule, than the direct reciprocity decision rule [20,21].”

23) 165-166 “the likelihood to allonurse should increase the reciprocity index values of pairs of mothers increases”: this sentence is grammatically incorrect. Also, the “reciprocity index values” have not yet been explained.

R23_RE3: We changed “reciprocity index values” for “as pairs of mothers are more reciprocal”. We wrote the following on lines 182-183: “2) the likelihood to allonurse should increase as pairs of mothers are more reciprocal.” We also replaced “the direct reciprocity index” with “the extent of reciprocity within pairs” on lines 186-187: “5) the extent of reciprocity within pairs of mothers should increase as the pairwise genetic relatedness of mothers increases.”

24) 170: Spello (reciporocity).

R24_RE3: We correct the spelling of reciprocity on lines 186-187: “5) the extent of reciprocity within pairs of mothers should increase as the pairwise genetic relatedness of mothers increases.”

25) 176: “subsequent to help given”.

R25_RE3: We did not change this and kept the following on lines 192-193: “To assess if reindeer mothers allonursed according to the generalized reciprocity decision rule, we asked if receiving help in general increased subsequent help given, …”

26) 179-180: Why? I would have predicted the opposite (because direct reciprocity should generate a higher propensity to provide help than generalised reciprocity; it is less prone to cheating).

R26_RE3: We agree with Reviewer 3 and wrote the following on lines 195-200: “We compared reciprocal allonursing in reindeer by both the generalized reciprocity and direct reciprocity decision rules. We propose that the propensity to help should be greater according to direct reciprocity than according to generalized reciprocity, since generalized reciprocity is less prone to cheating than direct reciprocity. Therefore, we predicted that 9) the latency to give help after receiving help should be shorter for direct reciprocity than generalized reciprocity.”

27) 458-459: “… did not significantly influence”

R27_RE3: We made the change on lines 483-485: “The conditional main effect for the pairwise genetic relatedness when the number of received help was equal to 0 did not significantly influence the number of help given …”

28) 470-471 and 492-493: I am afraid I do not understand how these three expected numbers are derived given the factor of increase is only 1.05 or 1.10, respectively. Why does the number of help of the receiver increase LESS (2.48 against 2.61) for each help received if the multiplication factor is higher (1.1 vs. 1.05)? Also, I would write “1” instead of “one” when referring to actual results.

R28_RE3: This is explained by the intercept values. Both results focus on the 2012 group. For the 2012 group with all the dyads, we wrote the following on lines 494-497: “For a one unit increase in the number of help received, the number help given is expected to increase by a factor of 1.05 (95% CI for IRR: 1.01 – 1.09). If a mother received help 1, 10 or 15 times, the mother is expected to give help 2.61, 4.10 and 5.26 times, respectively.” and the following on lines 502-503: “The intercept was significant (β ± SE: 0.91 ± 0.15, P < 0.001).” For the 2012 group with dyads with a tendency to be reciprocal, we wrote the following on lines 518-521: “For a one unit increase in the number of help received, the number help given is expected to increase by a factor of 1.10 (95% CI for IRR: 1.05 – 1.15). If a mother received help 1, 10 or 15 times, the mother is expected to give help 2.48, 5.58 and 8.76 times, respectively.” and the following on lines 526-527: “The intercept was significant (β ± SE: 0.82 ± 0.16, P < 0.001).” Thus, the estimated amount of help given by a mother is lower after one received help but is greater after multiple help received for the 2012 model with all dyads than for the 2012 with dyads with a tendency to reciprocate because of the estimated intercept and the estimated factor of increase. We also replaced “one” with “1” in both sentences on lines 496 and 520. 

29) 502-511: Perhaps for readability it would be good to start this long sentence with its main message that all these variables “did not significantly influence the number of help given in the 2013 closely-related group”.

R29_RE3: We revised the sentence as suggested on lines 571-579: “The number of help given in the 2013 closely-related group was not significantly influenced by the interaction between the number of help received and pairwise genetic relatedness (β ± SE: 0.72 ± 1.22, P = 0.56, IRR (95% CI): 2.05 (0.16 – 42.19)), the conditional main effect for the number of received help when pairwise genetic relatedness was equal to 0 (β ± SE: -0.06 ± 0.07, P = 0.41, IRR (95% CI): 0.94 (0.82 – 1.11)), the conditional main effect for the pairwise genetic relatedness when the number of received help was equal to 0 (β ± SE: -4.77 ± 4.10, P = 0.25, IRR (95% CI): 8.51e-3 (4.07e-7 – 23.41)), absolute rank difference (β ± SE: 0.07 ± 0.10, P = 0.51, IRR (95% CI): 1.07 (0.87 – 1.31)), similarity in offspring sex (β ± SE: 0.46 ± 0.28, P = 0.10, IRR (95% CI): 1.58 (0.84 – 2.75)), and offspring birth mass difference (β ± SE: 0.41 ± 0.28, P = 0.14, IRR (95% CI): 1.51 (0.78 – 2.66)).”

30) 515-516 and 527-528: “increased” seems redundant/ grammatically incorrect. Also, why is the potential dependency turned around here? Usually, you checked whether the help given is influenced by the help received, but here it seems opposite. If you look at this both ways, does this mean double testing the same data (even if from different ends)?

R30_RE3: We corrected this mistake and removed “increased”, which should not have been written in the text. We wrote the following on lines 584-586: “The number of help given was not significantly influenced by the number of help received (main effect: β ± SE: -0.05 ± 0.07, P = 0.49, IRR (95% CI): 0.96 (0.83 – 1.10)) in the 2013 closely-related group.” We did the same on lines 632-634: “The number of help given was not significantly influenced by the number of help received (main effect: β ± SE: 0.11 ± 0.09, P = 0.23, IRR (95% CI): 1.12 (0.92 – 1.34) in the 2013 distantly-related group.” 

31) 515-526 and 527-538: These two paragraphs seem to relate to the same dataset, “the 2013 distantly-related group”. What is the difference?

R31_RE3: This is a mistake that we should have noticed before submitting the last version. The first paragraph on lines 584-595 are the results for the 2013 closely-related group without the interaction between the number of help received and pairwise genetic relatedness. The second paragraph on lines 632-643 are the results for the 2013 distantly-related group without the interaction between the number of help received and pairwise genetic relatedness. We corrected this mistake on lines 584-586: “The number of help given was not significantly influenced by the number of help received (main effect: β ± SE: -0.05 ± 0.07, P = 0.49, IRR (95% CI): 0.96 (0.83 – 1.10)) in the 2013 closely-related group.”

32) 553-554, 582-583 and 608-609: I’m afraid this is tautological. RAFI is a measure of reciprocal allonursing, so a higher index value inadvertently means an “increased likelihood of allonursing”, doesn’t it?

R32_RE3: The reciprocity index, RAFI, is based on successful allonursing bouts given to each other by both mothers in a dyad. However, this index does not account for the allonursing rejections by mothers. The number of help received and the number of help given by mothers accounts for successful allonursing bouts received and given by mothers without accounting for the allonursing rejections by mothers. As such, we assessed whether more reciprocal dyads of mothers may be more likely to allonurse, i.e. the odds of a successful allonursing attempt may be greater than the odds of an unsuccessful allonursing attempt for more reciprocal dyads. If the likelihood of allonursing is influenced by RAFI, then this suggests that the odds of a successful allonursing attempt are greater for more reciprocal dyads, i.e. higher RAFI values, than the odds of an unsuccessful allonursing attempt. If there were no significant association between RAFI and the likelihood of allonursing, the odds of a successful allonursing attempt would not differ from the odds of an unsuccessful allonursing attempt, and this could happen if offspring attempted and were rejected just as often as they were accepted, independent of the dyadic reciprocity index. Since the odds of a successful allonursing attempt are greater than the odds of an unsuccessful allonursing attempt as RAFI increases, we view this as an important result that further supports the positive association between the amount of help received and the amount of help given by mother, which supports that reindeer mothers help partners according to the direct reciprocity decision rule.

We wrote the following in the methods on lines 393-399: “The number of help received and the number of help given by mothers accounts for successful allonursing bouts received and given by mothers without accounting for the allonursing rejections by mothers, i.e. unsuccessful allonursing attempts, however the likelihood of allonursing model accounts for both the successful and unsuccessful allonursing attempts. We assessed whether more reciprocal dyads of mothers may be more likely to allonurse, i.e. the odds of a successful allonursing attempt may be greater than the odds of an unsuccessful allonursing attempt for more reciprocal dyads.”

We wrote the following in the discussion on lines 703-706: “These results were further supported by the positive association between the dyadic index of reciprocity, RAFI, and likelihood of allonursing, since the odds of a successful allonursing attempt were greater than the odds of an unsuccessful allonursing attempt as RAFI increased in the 2012 group but not in the 2013 groups (prediction 3: main effect of RAFI on the likelihood of allonursing).”

We would be willing to move the likelihood of allonursing results to the Appendix, if Reviewer 3 and/or the Academic Editor would prefer that.

33) 554-555, 583-585 and 609-611: Without further specification this is redundant (pairwise genetic relatedness effects on the likelihood of allonursing were scrutinised already further above).

R33_RE3: Similar to our response, i.e. R32_RE3, the association between the number of allonursing bouts given by mothers and relatedness does not account for unsuccessful allonursing attempts, i.e. rejections, whereas the likelihood of allonursing does account for both successful and unsuccessful allonursing attempts. We found that the odds of a successful allonursing attempt did not differ from the odds of an unsuccessful allonursing attempt as relatedness between mother increased.

34) Fig. 2 seems redundant as explained by my last two comments.

R34_RE3: We hope to have convinced Reviewer 3 and the Academic Editor that these results are not redundant but rather interesting and important results. We would be willing to move the likelihood of allonursing results to the Appendix, if Reviewer 3 and/or the Academic Editor would prefer that.

35) 573 and 659: “did not significantly influence”

R35_RE3: We corrected these on lines 601 and 690.

36) 595-597 “The conditional main effect for RAFI when pairwise genetic relatedness was equal to 0 significantly influenced the likelihood to allonurse”: How can this be justified by a P-value of 0.74?

R36_RE3: We corrected this mistake and added “did not significantly influence the likelihood to allonurse” on lines 646-648: “The conditional main effect for RAFI when pairwise genetic relatedness was equal to 0 did not significantly influence the likelihood to allonurse (β ± SE: 0.46 ± 1.40, P = 0.74, OR (95% CI): 1.58 (0.10 – 24.63)).”

37) 640: “… as the offspring birth mass difference increased”

R37_RE3: We ran the model again to check the results and noticed the presence of multicollinearity. We had to delete the offspring birth mass difference and absolute rank difference of mothers from the model for multicollinearity values to be below 5. We report the results of the model on lines 671-677: “RAFI was not significantly influenced by pairwise genetic relatedness (main effect: β ± SE: 6.58 ± 47.82, P = 0.89, IRR (95% CI): 719.19 (1.41e-38 – 3.67e+43)) in the 2013 distantly-related group. The absolute difference in the age of mothers (β ± SE: 1.13 ± 1.02, P = 0.27, IRR (95% CI): 3.10 (0.42 – 22.89)) did not significantly influence RAFI. The intercept was marginally significant (β ± SE: -12.43 ± 6.64, P = 0.06). The random intercept effect for the first mother in a dyad explained 12.41 (SD = 3.52) of the variance, and the random intercept effect for the other mother in a dyad explained 126.31 (SD = 11.24) of the variance. The model’s conditional R2 was equal to 1.00.” 

38) 677-678 “A positive association between help given and help received within dyads is a commonly accepted result for allogrooming by direct reciprocity in primates”: This applies also to Norway rats (Stieger et al. 2017, Behavioral Ecology and Sociobiology 71: 182).

R38_RE3: We agree with Reviewer 3 and wrote the following on lines 709-712: “A positive association between help given and help received within dyads is a commonly accepted result for allogrooming by direct reciprocity in primates [62,140,163], for food donations by direct reciprocity in Norway rats [61,63,71,164–168] and for food sharing by direct reciprocity in vampire bats [57].”

39) 679 “… and for food sharing by direct reciprocity in vampire bats”: And Norway rats (ref. 45).

R39_RE3: We added Kettler et al. (2021). Please also see our response R38_RE3. 

40) 686: What is “a different evolved explanation for cooperation”?

R40_RE3: We revised this sentence to refer to the mechanism of cooperation of by-product mutualism, enforcement and deception. We wrote the following on lines 716-731: “The question of “why was there no correlational evidence of reindeer mothers in both 2013 groups cooperating with social partners according to the direct reciprocity decision rule?” is unlikely to be explained by the cooperation mechanism of i) by-product mutualism [25,31–34], which cannot be cheated, since allonursing can be cheated by offspring stealing milk [7,106,169], and ii) enforcement [40,41], since neither the mothers nor the offspring coerced the other to transfer milk from the mother to the offspring. Allonursing can be a product of offspring stealing milk [6,7], and reindeer offspring steal milk [106]. Thus, milk parasitism as a form of cooperation by deception plays a role in this communal breeding social system. Nonetheless, additional observation days were very likely required to assess the direct reciprocity decision rule in 2013. The difference between years is the time period of observation, i.e. 65 observation days in 2012 versus 25 observation days in 2013. The latency to give help after receiving help according to the direct reciprocity decision rule for allonursing reindeer mothers is on average 19.42 ± 0.45 days, yet the observation period spans 25 observation days in 2013. Returning allonursing help received was not observed to occur simultaneous, within a few minutes and within a day because i) lactation is the most energetically costly aspect of mammalian biology [8,9], and ii) reindeer have smaller udders than most ungulates [119].”

41) 688: “give” (not “given”).

R41_RE3: We corrected this on lines 726-727: “The latency to give help after receiving help …”

42) 720-721: The logic of correlated pay-offs is usually applied to the effects of cooperation on the partners involved; their pay-offs may be correlated by a reciprocal exchange of service in iterated interactions, or by sharing genes (cf. ref. 21). I am not aware of an expectation that “direct reciprocity and relatedness” should “have correlated pay-offs”.

R42_RE3: We deleted the sentence.

43) 730-731: There is some grammatical problem in this sentence; as such it does not make sense.

R43_RE3: We revised the sentence on lines 764-765: “A meta-analysis found that the incidence of allonursing was not associated with relatedness in groups where females associate with kin [15].”

44) 755: “… according to one”

R44_RE3: We included “according to one” in the sentence on lines 790-792: “A future experimental design is needed to assess if reindeer mothers allonurse according to generalized reciprocity and direct reciprocity decision rules or only according to one of these two decision rules.”

45) 755: I think ref. 32 should be mentioned here as well.

R45_RE3: We added Axelrod and Hamilton (1981) to the following on lines 792-794: “Both the direct reciprocity [28,35,36] and the generalized reciprocity [48,49,73,108,110] decision rules can lead to the evolution of cooperation.”

46) 756: Ref. 30 would seem adequate here as well, and perhaps also Rankin & Taborsky 2009 (Evolution 63, 1913-1922) because it modelled the evolution of generalised reciprocity under most general conditions.

R46_RE3: We added both van Doorn and Taborsky (2012) and Rankin and Taborsky (2009) on lines 792-794: “Both the direct reciprocity [28,35,36] and the generalized reciprocity [48,49,73,108,110] decision rules can lead to the evolution of cooperation.”

47) 758: I suggest writing something like “… and information about whether specific individuals …”

R47_RE3: We made the suggested change on lines 794-796: “Generalized reciprocity is cognitively less demanding than direct reciprocity, and generalized reciprocity does not require individual recognition and information about whether specific individuals previously helped them [21].”

48) 765-766: The study in longtailed macaques was not well suited test for generalised reciprocity because spatial proximity was not controlled for (cf. p. 151 in ref. 20).

R48_RE3: We wrote the following on lines 801-804: “Longtailed macaques help according to direct reciprocity and indirect reciprocity, i.e. help someone who is helpful (if A helped B, C helps A) but not according to generalized reciprocity, however spatial proximity may have affected the assessment of the generalized reciprocity decision rule [72].”

49) 769-770: There were some more experimental studies, e.g. Schneeberger et al. 2012 (BMC Evolutionary Biology 2012, 12:41) and Gerber et al. 2020 (Proc. R. Soc. B 287: 20202327).

R49_RE3: We included these two studies on lines 806-808: “The generalized reciprocity decision rule may be widespread and should be investigated in both correlational and experimental studies, yet there are few empirical studies of the generalized reciprocity decision rule in animals [60,66,68–70,72,164,165].”

50) 778: Can you cite evidence for the herders’ observations?

R50_RE3: We wrote the following on lines 816-818: “Sámi reindeer herders observe allonursing in large herds of semi-domesticated reindeer (personal communication with Sámi reindeer herders at the Sámi Education Institute in Inari and Kaamanen, Finland).”

51) 798: How can a “study” be “tested”?

R51_RE3: We wrote the following on lines 837-838: “The alloparental care literature has rarely assessed if individuals help partners according to the direct reciprocity decision rule.”

52) 801: “Roulin 2002” does not comply with the usual citation format.

R52_RE3: We revised the citation format on lines 839-841: “Furthermore, there is yet no evidence to support that two females achieve a higher fitness when allonursing reciprocally than when they do not [7].”

53) 803-805 “a long time delay does not exclude the possibility…”: I would rather suggest it “hints on this possibility”. Without this long-term memory, these results could not be easily explained.

R53_RE3: We revised the sentence as follows on lines 842-845: “The latency to give help after receiving help extended to several days, i.e. on average 19.42 days, by direct reciprocity, however a long time delay hints to the possibility that reindeer may be capable of individual recognition, the ability to remember whether individuals previously helped them and the outcomes of previous encounters.”

54) 939: This reference (47) needs to be corrected.

R54_RE3: We corrected the reference on lines 997-998: “63. Engelhardt SC, Taborsky, M. Food-exchanging Norway rats apply the direct reciprocity decision rule rather than copying by imitation. Anim Behav. 2022;194: 265-274.”

Michael Taborsky

(I always sign my reviews because I am in favour of transparency in science; cf. Ethology 113 (2007), 1–8)

7. PLOS authors have the option to publish the peer review history of their article (what does this mean?). If published, this will include your full peer review and any attached files.

Do you want your identity to be public for this peer review? For information about this choice, including consent withdrawal, please see our Privacy Policy.

Reviewer #1: No

Reviewer #3: Yes: Michael Taborsky

R2_AE: The attached files were for the first round of review. There are no attached files for the second round of review.

R3_AE: We did use PACE previously, and the Production Editor Glenn Jackson confirmed that these met the requirements for PLOS ONE figures.

---

## [Decision Letter · Decision Letter 2]

5 Oct 2023

PONE-D-22-28248R2Evidence suggesting that reindeer mothers allonurse according to the direct reciprocity and generalized reciprocity decision rulesPLOS ONE

Dear Dr. Engelhardt,

Thank you for submitting your manuscript to PLOS ONE. After careful consideration, we feel that it has merit but does not fully meet PLOS ONE’s publication criteria as it currently stands. Therefore, we invite you to submit a revised version of the manuscript that addresses the points raised during the review process.

I’ve just taken over as Editor on this submission because the previous Editor was unable to continue. Both Reviewers have now recommended Acceptance (with very minor changes), but on my reading of the manuscript, there are a couple of issues that need to be addressed before publication. I don’t want to ask for substantial changes at this late stage, but since one reviewer is demanding some minor changes anyway, you should attend to the following points in your revision.

First, the methods makes it sound like the reindeer were removed from their mothers, or that the mother-infant pairs were removed from the rest of the herd. If so, this would remove any contextual cues of who their kin were. Many mammals rely on such contextual cues like “who did I grow up with”, “who else does my mother care for”, so removing the animals would remove their ability to distinguish kin from non-kin. By analogy, it would be like removing a human infant (or non-verbal mother-infant pair) from their family, then testing many years later whether that child would be more generous towards its siblings without telling it who the siblings were. Back to this manuscript, the authors need to clarify in the manuscript who was removed and when. If the allonursers were removed when they themselves were young, then they would have had no contextual cues of kinship, so it is unsurprising that they didn’t allonurse more for kin. In this case, the authors would need to tone down all their conclusions about kinship, discuss how the reindeer didn’t have any contextual cues to help them discriminate kin, starting right from the abstract with a statement like “though this is possibly because they lacked contextual kinship cues”. If instead it was the allonurser’s offspring who were removed, then the authors need to discuss the consequences of this. For example, would it result in more allnursing than normal, given that the females didn’t have their own offspring to invest in?

Second, in the discussion, it sounds like the 2013 meta-analysis is underpowered. Normally, I’d recommend that you conduct an internal meta-analysis of the 2012 and 2013 data to check the overall effects – this would be the best way to report these data. However, given how late it is in the review process, I’m not going to demand such a major overhaul. Instead, you should do one of the following: a) do a quick power analysis to demonstrate that 2013 does indeed have sufficient power; b) acknowledge everywhere (including up front & in the abstract) that the 2013 null results could be due to insufficient power; c) do the internal meta-analysis of 2012 & 2013 I suggested and adjust the discussion accordingly (this would improve the presentation of the results).

Third, the first few pages of the manuscript are challenging to read. The sentences within the paragraphs seem to jump around in a disconnected way. Many paragraphs are also very long (>1 page), and contain multiple topics that would be better separated into different paragraphs. This isn’t the strongest way to start, and will lose some readers. Ideally, you would think hard about the information in those paragraphs and reorganize the information into smaller packages. However, this late in the review process, I’ll settle for just breaking up the long paragraphs into smaller paragraphs – aim for no more than ½ to ¾ page, though shorter is fine too. Doing so will improve your paper’s readability, and thus its impact.

Once you have addressed these concerns and the final concerns of the one reviewer, then the paper should be acceptable for publication.

We look forward to receiving your revised manuscript.

Kind regards,

Pat Barclay

Academic Editor

PLOS ONE

Journal Requirements:

Reviewers' comments:

Reviewer's Responses to Questions

**Comments to the Author**

1. If the authors have adequately addressed your comments raised in a previous round of review and you feel that this manuscript is now acceptable for publication, you may indicate that here to bypass the “Comments to the Author” section, enter your conflict of interest statement in the “Confidential to Editor” section, and submit your "Accept" recommendation.

Reviewer #1: All comments have been addressed

Reviewer #3: (No Response)

2. Is the manuscript technically sound, and do the data support the conclusions?

Reviewer #1: Partly

Reviewer #3: Yes

3. Has the statistical analysis been performed appropriately and rigorously? 

Reviewer #1: Yes

Reviewer #3: Yes

4. Have the authors made all data underlying the findings in their manuscript fully available?

Reviewer #1: Yes

Reviewer #3: (No Response)

5. Is the manuscript presented in an intelligible fashion and written in standard English?

Reviewer #1: Yes

Reviewer #3: Yes

6. Review Comments to the Author

Reviewer #1: (No Response)

Reviewer #3: I commend the authors for revising their manuscript satisfactorily. Most of problems I have indicated have been removed. There are just a few issues left as outlined in my detailed comments below.

Detailed comments (by line numbers):

39-40: I think this needs to be amended otherwise it is not clear whether it refers to direct or generalised reciprocity (e.g. by saying something like: “i) increased as the number of help received FROM THE SAME PARTNER increased”

41-42: Before the sentence “The overall number of help given increased as the overall number of help received increased” I would add “iv)”.

195-200 (line number given in response letter): I’m afraid this is a misunderstanding. In my previous comment “direct reciprocity should generate a higher propensity to provide help than generalised reciprocity; it is less prone to cheating” I meant that DIRECT reciprocity is less prone to cheating, because the behaviour of interacting partners is directly controlled among them by their mutual responses; some researchers therefore allude to reciprocity as being “enforced”. In generalised reciprocity cheating is not directly prevented by punishment but rather indirectly causes a decline in mutual cooperation among interacting partners in a community. Hence I personally would switch the argument around and write something like: “… since DIRECT reciprocity is less prone to cheating than GENERALISED reciprocity. Therefore, EFFECTS OF RECEIVED HELP SHOULD VANISH MORE QUICKLY IF GENERALISED RECIPROCITY APPLIES, SO we predicted that 9) the DISTRIBUTION OF latencIES to give help after receiving help should PEAK AT shorter TIME INTERVALS for GENERALISED reciprocity than FOR DIRECT reciprocity.”

709-712 (line number given in response letter): With my previous comment on this sentence I wanted to point out that also ALLOGROOMING in Norway rats showed “a positive association between help given and help received” (Stieger et al. 2017, Behav Ecol Sociolbiol 71: 182).

781-783 (line number given in response letter): Here you should first stress that older mothers respond quicker to received help by returning it after shorter latency periods, and then you should offer potential explanations (e.g. that older mothers are more experienced in reciprocal interactions than younger individuals). What you now state is just a more elaborated description of the result, not an interpretation.

813: For clarity I propose to write “and reindeer offspring DO steal milk”

819-820: “… not observed to occur simultaneousLY"

Michael Taborsky

(I always sign my reviews because I am in favour of transparency in science; cf. Ethology 113 (2007), 1–8)

7. PLOS authors have the option to publish the peer review history of their article (what does this mean?). If published, this will include your full peer review and any attached files.

Reviewer #1: No

Reviewer #3: **Yes: **Michael Taborsky

---

## [Author Response · Author response to Decision Letter 2]

3 Nov 2023

PONE-D-22-28248R2

Evidence suggesting that reindeer mothers allonurse according to the direct reciprocity and generalized reciprocity decision rules

PLOS ONE

Dear Dr. Engelhardt,

Thank you for submitting your manuscript to PLOS ONE. After careful consideration, we feel that it has merit but does not fully meet PLOS ONE’s publication criteria as it currently stands. Therefore, we invite you to submit a revised version of the manuscript that addresses the points raised during the review process.

I’ve just taken over as Editor on this submission because the previous Editor was unable to continue. Both Reviewers have now recommended Acceptance (with very minor changes), but on my reading of the manuscript, there are a couple of issues that need to be addressed before publication. I don’t want to ask for substantial changes at this late stage, but since one reviewer is demanding some minor changes anyway, you should attend to the following points in your revision.

AE_1: First, the methods makes it sound like the reindeer were removed from their mothers, or that the mother-infant pairs were removed from the rest of the herd. If so, this would remove any contextual cues of who their kin were. Many mammals rely on such contextual cues like “who did I grow up with”, “who else does my mother care for”, so removing the animals would remove their ability to distinguish kin from non-kin. By analogy, it would be like removing a human infant (or non-verbal mother-infant pair) from their family, then testing many years later whether that child would be more generous towards its siblings without telling it who the siblings were. Back to this manuscript, the authors need to clarify in the manuscript who was removed and when. If the allonursers were removed when they themselves were young, then they would have had no contextual cues of kinship, so it is unsurprising that they didn’t allonurse more for kin. In this case, the authors would need to tone down all their conclusions about kinship, discuss how the reindeer didn’t have any contextual cues to help them discriminate kin, starting right from the abstract with a statement like “though this is possibly because they lacked contextual kinship cues”. If instead it was the allonurser’s offspring who were removed, then the authors need to discuss the consequences of this. For example, would it result in more allnursing than normal, given that the females didn’t have their own offspring to invest in?

Re_AE_1: Thank you for this comment. This is a misunderstanding, and we revised the text to help clarify. This study is from the point of the view of the mothers, and we investigated if the mothers allonursed according to three different decision rules. To help clarify, we revised the Methods section on lines 228-234: “No mother was removed from their own mother when they themselves were young. Mothers were not separated from their offspring. Thus, reindeer mothers’ ability to distinguish kin from non-kin was not affected. Mothers gave birth to their offspring and raised their off-spring with the rest of the herd in the calving paddocks for two to 5 weeks. Mother-offspring pairs selected for the study were then separated from the rest of the herd, which was re-leased in a large calving ground area. The study animals and the herd were rejoined at the end of the study.” 

AE_2: Second, in the discussion, it sounds like the 2013 meta-analysis is underpowered. Normally, I’d recommend that you conduct an internal meta-analysis of the 2012 and 2013 data to check the overall effects – this would be the best way to report these data. However, given how late it is in the review process, I’m not going to demand such a major overhaul. Instead, you should do one of the following: a) do a quick power analysis to demonstrate that 2013 does indeed have sufficient power; b) acknowledge everywhere (including up front & in the abstract) that the 2013 null results could be due to insufficient power; c) do the internal meta-analysis of 2012 & 2013 I suggested and adjust the discussion accordingly (this would improve the presentation of the results).

Re_AE_2: Thank you for bringing up this important point. As the editor suggested, we ran two meta-analyses, and both were significant. We ran one meta-analysis with the incidence rate ratio as the effect size based on the generalized linear mixed models with a Poisson distribution for the association between the amount of help given and received. We ran a second meta-analysis with the odds ratio as the effect size based on the generalized linear mixed models with a binomial distribution with the likelihood of allonursing as the response variable. 

We wrote the following on lines 422-428: “Two internal meta-analyses were performed to assess the direct reciprocity decision rule among the three groups of reindeer mothers with the inverse variance method of pooling to assess the pooled effect size with fixed effects models based on the between-study heterogeneity. The effect sizes were the incidence rate ratio and the odds ratio, and both effect sizes were derived from the generalized linear mixed models. The estimates of the effect sizes were comparable among groups, since they were generated from generalized linear mixed models with the same distributions, and fixed and random effects.”

We wrote the following in the Results section on lines 696-711: “Meta-analyses for the direct reciprocity decision rule

 For each additional help received by reindeer mothers, the rate of help given increased by 1.05 (fixed effects model for the incidence rate ratio: pooled effect size = 1.05 (95% CI: 1.01 – 1.08), p < 0.001, N = 3 effect sizes), which supports the direct reciprocity decision rule for the amount of help given and received. All effect sizes shared the same true effect size (Q = 2.00, df = 2, p = 0.37; tau2 = 0.0001 (95% CI: 0.0000 – 0.2460); I2 = 0.1% (95% CI: 0.0% – 89.6%); H = 1.00 (95% CI: 1.00 – 3.10)). The effect size for the 2012 group accounted for 89.7% of the weight for the pooled effect size for the incidence rate ratio, whereas the 2013 closely-related and distantly-related groups accounted for 7.3% and 3.0%, respectively, of the weight for the pooled effect size. The odds of allonursing bout increased as RAFI increased (fixed effects model for OR: pooled effect size = 1.81 (95% CI: 1.38 – 2.39), p < 0.001, N = 3 effect sizes), which supports the direct reciprocity decision rule for the likelihood of a successful allonursing bout. All effect sizes shared the same true effect size (Q = 5.32, df = 2, p = 0.07; tau2 = 0.27 (95% CI: 0.00 – 12.36); I2 = 62.4% (95% CI: 0.0% – 89.3%); H = 1.63 (95% CI: 1.00 – 3.05)). The effect size for the 2012 group accounted for 87.0% of the weight for the pooled effect size for the OR, whereas the 2013 closely-related and distantly-related groups accounted for 10.1% and 2.8%, respectively, of the weight for the pooled effect size.”

We wrote the following in the Discussion section on lines 748-752: “The internal meta-analysis results further supported the direct reciprocity decision rule among the three groups of reindeer mothers in 2012 and 2013, since 1) the rate of help given increased as help received increased, and 2) the odds of allonursing increased as RAFI increased among the 3 groups. The weight of the 2012 group for the pooled effect sizes was ≥ 87.0, which explains the significant internal meta-analysis results among the three groups.” 

With the addition, we re-read the Discussion several times. We think that the following paragraph remains important for Discussion about the lack of a positive association in the 2013 groups on lines 756-771: “The question of “why was there no correlational evidence of reindeer mothers in both 2013 groups cooperating with social partners according to the direct reciprocity decision rule?” is unlikely to be explained by the cooperation mechanism of i) by-product mutualism [25,31–34], which cannot be cheated, since allonursing can be cheated by offspring stealing milk [7,108,171], and ii) enforcement [40,41], since neither the mothers nor the offspring coerced the other to transfer milk from the mother to the offspring. Allonursing can be a product of offspring stealing milk [6,7], and reindeer offspring steal milk [108]. Thus, milk parasitism as a form of cooperation by deception plays a role in this communal breeding social system. Nonetheless, additional observation days were very likely required to assess the direct reciprocity decision rule in 2013. The difference between years is the time period of observation, i.e. 65 observation days in 2012 versus 25 observation days in 2013. The latency to give help after receiving help according to the direct reciprocity decision rule for allonursing reindeer mothers is on average 19.42 ± 0.45 days, yet the observation period spans 25 observation days in 2013. Returning allonursing help received was not observed to occur simultaneously, within a few minutes and within a day because i) lactation is the most energetically costly aspect of mammalian biology [8,9], and ii) reindeer have smaller udders than most ungulates [121].” 

The effect sizes for each group were already reported with the model results. We added the codes to the Rscript.

AE_3: Third, the first few pages of the manuscript are challenging to read. The sentences within the paragraphs seem to jump around in a disconnected way. Many paragraphs are also very long (>1 page), and contain multiple topics that would be better separated into different paragraphs. This isn’t the strongest way to start, and will lose some readers. Ideally, you would think hard about the information in those paragraphs and reorganize the information into smaller packages. However, this late in the review process, I’ll settle for just breaking up the long paragraphs into smaller paragraphs – aim for no more than ½ to ¾ page, though shorter is fine too. Doing so will improve your paper’s readability, and thus its impact.

Re_AE_3: Thank you for this comment. We revised the introduction and separated paragraphs that contained multiple topics. 

Comment for the Academic Editor: Please note that the track changes on lines 532-534 with the field code changes and the Spanish (latin america) formatting are odd to me. I keep trying to reject these, and Word keeps bringing this up again and again. I assume this is an issue with the Word/Mendeley plug-in. Please ignore these track changes.

Once you have addressed these concerns and the final concerns of the one reviewer, then the paper should be acceptable for publication.

We look forward to receiving your revised manuscript.

Kind regards,

Pat Barclay

Academic Editor

PLOS ONE

Journal Requirements:

Please note that we added two references, which were recently accepted. One is already published (Engelhardt and Taborsky 2023), and the other paper is in the proofs stage (Engelhardt and Taborsky Accepted in Ethology). We added the reference that reviewer 3 commented on, i.e. Stieger et al. 2017, Behav Ecol Sociolbiol 71: 182. We removed McPherson et al. (2001) and Schürch et al. (2010) from the references: they were not cited. We removed two additional citations and moved the order of two citations. These changes affected the numbering of the references and the citations. 

Reviewers' comments:

Reviewer's Responses to Questions

Comments to the Author

1. If the authors have adequately addressed your comments raised in a previous round of review and you feel that this manuscript is now acceptable for publication, you may indicate that here to bypass the “Comments to the Author” section, enter your conflict of interest statement in the “Confidential to Editor” section, and submit your "Accept" recommendation.

Reviewer #1: All comments have been addressed

Reviewer #3: (No Response)

2. Is the manuscript technically sound, and do the data support the conclusions?

Reviewer #1: Partly

Reviewer #3: Yes

3. Has the statistical analysis been performed appropriately and rigorously? 

Reviewer #1: Yes

Reviewer #3: Yes

4. Have the authors made all data underlying the findings in their manuscript fully available?

Reviewer #1: Yes

Reviewer #3: (No Response)

5. Is the manuscript presented in an intelligible fashion and written in standard English?

Reviewer #1: Yes

Reviewer #3: Yes

6. Review Comments to the Author

Reviewer #1: (No Response)

Thank you for your comments and effort.

Reviewer #3: I commend the authors for revising their manuscript satisfactorily. Most of problems I have indicated have been removed. There are just a few issues left as outlined in my detailed comments below.

Thank you for your comments.

Detailed comments (by line numbers):

R3_1: 39-40: I think this needs to be amended otherwise it is not clear whether it refers to direct or generalised reciprocity (e.g. by saying something like: “i) increased as the number of help received FROM THE SAME PARTNER increased”

Re_R3_1: We revised the sentence to include “from the same partner” on lines 39-42: “The number of help given i) increased as the number of help received from the same partner increased in the 2012 group but not in both 2013 groups, ii) was not influenced by relatedness, and iii) was not influenced by an interaction between the number of help received from the same partner and relatedness.”

R3_2: 41-42: Before the sentence “The overall number of help given increased as the overall number of help received increased” I would add “iv)”.

Re_R3_2: We added “iv)” before the sentence on lines 42-43: “iv) The overall number of help given increased as the overall number of help received increased.”

R3_3: 195-200 (line number given in response letter): I’m afraid this is a misunderstanding. In my previous comment “direct reciprocity should generate a higher propensity to provide help than generalised reciprocity; it is less prone to cheating” I meant that DIRECT reciprocity is less prone to cheating, because the behaviour of interacting partners is directly controlled among them by their mutual responses; some researchers therefore allude to reciprocity as being “enforced”. In generalised reciprocity cheating is not directly prevented by punishment but rather indirectly causes a decline in mutual cooperation among interacting partners in a community. Hence I personally would switch the argument around and write something like: “… since DIRECT reciprocity is less prone to cheating than GENERALISED reciprocity. Therefore, EFFECTS OF RECEIVED HELP SHOULD VANISH MORE QUICKLY IF GENERALISED RECIPROCITY APPLIES, SO we predicted that 9) the DISTRIBUTION OF latencIES to give help after receiving help should PEAK AT shorter TIME INTERVALS for GENERALISED reciprocity than FOR DIRECT reciprocity.”

Re_R3_3: We revised the sentences as suggested on lines 200-205: “We propose that the propensity to help should be greater according to direct reciprocity than according to generalized reciprocity, since direct reciprocity is less prone to cheating than generalized reciprocity. Therefore, the effects of received help should vanish more quickly if generalized reciprocity applies, so we predicted that 9) the distribution of latencies to give help after receiving help should peak at shorter time intervals for generalized reciprocity than for direct reciprocity.”

R3_4: 709-712 (line number given in response letter): With my previous comment on this sentence I wanted to point out that also ALLOGROOMING in Norway rats showed “a positive association between help given and help received” (Stieger et al. 2017, Behav Ecol Sociolbiol 71: 182).

Re_R3_4: We added Stieger et al. (2017) for allogrooming in Norway rats on lines 743-745: “A positive association between help given and help received within dyads is a commonly accepted result for allogrooming by direct reciprocity in primates [62,142,165] and in Norway rats [166] …”

R3_5: 781-783 (line number given in response letter): Here you should first stress that older mothers respond quicker to received help by returning it after shorter latency periods, and then you should offer potential explanations (e.g. that older mothers are more experienced in reciprocal interactions than younger individuals). What you now state is just a more elaborated description of the result, not an interpretation.

Re_R3_5: We revised the sentence as follows on lines 821-823: “Older mothers returned received help sooner than younger mothers, and we suggest that older mothers may be more experienced in reciprocal interactions than younger mothers.”

R3_6: 813: For clarity I propose to write “and reindeer offspring DO steal milk”

Re_R3_6: We added “do” on lines 761-762: “and reindeer offspring do steal milk [108].”

R3_7: 819-820: “… not observed to occur simultaneousLY"

Re_R3_7: We corrected this on lines 768-769: “… not observed to occur simultaneously".

Michael Taborsky

(I always sign my reviews because I am in favour of transparency in science; cf. Ethology 113 (2007), 1–8)

7. PLOS authors have the option to publish the peer review history of their article (what does this mean?). If published, this will include your full peer review and any attached files.

Do you want your identity to be public for this peer review? For information about this choice, including consent withdrawal, please see our Privacy Policy.

Reviewer #1: No

Reviewer #3: Yes: Michael Taborsky

---

## [Editor Report · Decision Letter 3]

24 Nov 2023

Evidence suggesting that reindeer mothers allonurse according to the direct reciprocity and generalized reciprocity decision rules

PONE-D-22-28248R3

Dear Dr. Engelhardt,

We’re pleased to inform you that your manuscript has been judged scientifically suitable for publication and will be formally accepted for publication once it meets all outstanding technical requirements.

Kind regards,

Pat Barclay

Academic Editor

PLOS ONE
---

## [Editor Report · Acceptance letter]

5 Dec 2023

PONE-D-22-28248R3 

Evidence suggesting that reindeer mothers allonurse according to the direct reciprocity and generalized reciprocity decision rules 

Dear Dr. Engelhardt:

I'm pleased to inform you that your manuscript has been deemed suitable for publication in PLOS ONE. Congratulations! Your manuscript is now with our production department. 

Kind regards, 

on behalf of

Dr. Pat Barclay 

Academic Editor

PLOS ONE